# Decadal Changes in the Leading Patterns of Sea Level Pressure in the Arctic and Their Impacts on the Sea Ice Variability in Boreal Summer

Nakbin Choi[1], Kyu-Myong Kim[2], Young-Kwon Lim[3] and Myong-In Lee[1]

[1]School of Urban & Environmental Engineering, Ulsan National Institute of Science and Technology, Ulsan, Republic of Korea
[2]Climate and Radiation Laboratory, NASA, Goddard Space Flight Center, MD, USA
[3]Global Modeling and Assimilation Office, NASA Goddard Space Flight Center, MD, USA

*Correspondence to*: Myong-In Lee (milee@unist.ac.kr)

**Abstract.** Besides its negative trend, the interannual and the interdecadal changes in the Arctic sea ice are also pronounced in recent decades. The three leading modes in the sea level pressure (SLP) variability in the Arctic (70°–90°N) – the Arctic Oscillation (AO), the Arctic Dipole (AD), and the third mode (A3) – are analyzed to understand the linkage between sea ice variability and large-scale atmospheric circulation in boreal summer (June-August). This study also compares the decadal changes of the modes between the early (1982-1997) and the recent (1998-2017) periods and their influences on the Arctic sea ice extent (SIE).

Only the AD mode shows a significant correlation increase with SIE in summer (JJA) from -0.05 in the early period to 0.57 in the recent period. The AO and the A3 modes show a less significant relationship with SIE for the two periods. The AD is characterized by a dipole pattern of SLP, which modulates the strength of meridional surface winds and the transpolar drift stream (TDS). The major circulation change in the late 1990s is that the direction of the wind has been changed more meridionally over the exit region of the Fram Strait, which causes sea ice drift and discharge through that region. In addition, the response of surface albedo and the net surface heat flux becomes larger and much clearer, suggesting a positive sea ice-albedo feedback in the sea ice variability associated with the AD. The analysis also reveals that the zonal shift of the centers of SLP anomalies and associated circulation change affects a significant reduction in sea ice concentration over the Pacific sector of the Arctic Ocean. This study further suggests that the Pacific Decadal Oscillation (PDO) phase change could influence the spatial pattern change in the AD.

## 1 Introduction

Arctic sea ice has experienced a radical change recently with the record-breaking minimum in the sea ice extent (SIE) in 2012 (Parkinson and Comiso, 2013). Based on the National Snow and Ice Data Center (NSIDC, 2019a), the linear trend of the SIE during 1979-2018 relative to 1981-2010 average is -12.8 % per decade, with a more rapid declining trend in recent years. Sea ice melting is obviously linked to global warming-induced by increasing greenhouse gases. It is also suggested that the sea ice

variability in the Arctic could affect the weather and climate extremes in the mid-latitudes such as the occurrence frequency of summertime heatwaves (Tang et al., 2014)

Beside its downward trend, the Arctic SIE also exhibits a pronounced interannual and interdecadal change, which is especially prominent in its minimum phase in September. Underlying mechanisms for the long-term sea ice variability in summer from interannual to interdecadal time scale have been discussed based on a variety of mechanisms in previous literature. The variation of the Arctic SIE is affected locally by shortwave and longwave radiation during summer. For example, Curry et al. (1995) suggested the sea ice-albedo feedback in accelerating sea ice melting and global warming. Since satellite observations available, Parkinson et al. (1999) analyzed the Arctic SIE variability and its trends. Recently, Wernli and Papritz (2018) emphasized the role of the synoptic weather pattern in the sea ice variability such as the Arctic anticyclones with a typical duration of 10 days. They showed a significant correlation between the seasonal reduction of the Arctic sea ice and the occurrence frequency of anticyclones poleward of 70° N. Warm air advection and/or oceanic heat transport may also have significant impacts on the interannual and the interdecadal variability of SIE. For example, a strengthened upper ocean circulation can enhance oceanic heat transport either from the Pacific (Shimada et al., 2006) and/or from the Atlantic (Polyakove et al., 2005), thereby melting sea ice drastically.

Other possible processes that have drawn less attention in the previous studies are the changes in the dynamical drift of sea ice driven by atmospheric and oceanic circulation. Thorndike and Colony (1982) indicated that both atmospheric and oceanic circulations were equally important in driving long-term sea ice motion in the central Arctic in the time scale averaged for several months, while the atmospheric circulation was more important in shorter time scales and more than 70 % variance of sea ice velocity was related to the geostrophic wind. Ogi et al. (2010) suggested the important role of surface wind change in the interannual variation of the Arctic SIE in summer. Park and Stewart (2016) also analyzed the role of surface wind for sea ice drift using an analytical model, and demonstrated that strengthening southerly wind could effectively decrease summertime sea ice over the Pacific sector of the Arctic. The Transpolar Drift Stream (TDS) is known to be important for the sea ice outflow from the Arctic Ocean (NSIDC, 2019b). This process was validated in a modeling study of Wettstein and Deser (2014), who investigated the relationship between the Arctic sea ice loss and the large-scale atmospheric circulation using the Community Climate System Model version 3 (CCSM3) experiments with large ensemble members during the period of 2000-2061. In their study, a higher loss rate of the summertime Arctic sea ice was engaged with a more enhanced transpolar drift induced by large-scale atmospheric circulation pattern in the model. They suggested a similar mechanism is likely to work in the interannual timescale.

The Arctic Oscillation (AO) is known as the most dominant mode of atmospheric circulation over the northern hemisphere (Thompson and Wallace, 1998). The second mode is the east-west dipole mode, which has been known with different names in previous literatures, such as the Dipole Anomaly (Wu et al. 2006; Watanabe et al. 2006), Barents Oscillation (Skeie, 2000), or the Arctic Dipole (Overland and Wang, 2010; AD hereafter). Despite some studies regarded this mode as an unrealistic one that statistically produced by the temporary shift of AO in the EOF analysis (e.g., Tremblay, 2001), there have been many other studies suggesting this mode is physically robust in the Arctic (e.g., Overland and Wang 2005; Chen et al., 2013). In

particular, Overland and Wang (2005) showed that the AD mode is evident as the third EOF mode in sea level pressure (SLP) in the Northern Hemisphere mid-latitudes over 20°N. This mode is associated with a meridional wind dominantly, which tends to affect the meridional sea ice transport from the Arctic to the Atlantic in boreal winter (Wu et al., 2006; Watanabe et al., 2006). These circulation mode changes affect the Arctic sea ice distribution not only in winter but also in summer, and both

the AO and AD modes were linked with the September SIE variability in the previous studies. Ogi and Wallace (2007) showed that the SLP field regressed onto the timeseries of the September SIE was projected as the AO pattern in summer, and Ogi et al. (2008) suggested that the surface wind anomalies associated with AO are mainly responsible for the sea ice loss in summer. Wang et al. (2009) identified AO and AD as the two principal modes of SLP from the analysis of the long-term data for 1948-2008 and suggested that negative AD years such as 2007 tend to show more linkage with the SIE minimum. Overland and

Wang (2010) focused on the rapid melting of sea ice in summer driven by large-scale atmospheric circulation changes, especially by AO and AD. Ballinger and Rogers (2014) suggested AD is one of the important modes in driving sea ice variability in the western Arctic.

In addition to the interannual variability of SIE, the mechanisms behind the decadal change need to be understood in more detail to explain the rapid melting of sea ice in the recent decade. Many recent studies have highlighted the decadal change in

large-scale atmospheric circulation in the numerous regions of the globe after the mid-1990s. For example, Kwon et al. (2005) and Yim et al. (2008) indicated a significant change in the Asian summer monsoon rainfall after the mid-1990s. Kang et al. (2014) also showed the increase of the potential predictability of AO through the enhanced linkage between tropical El Nino and Southern Oscillation (ENSO) and AO during winter after the late 1990s. Not only the atmospheric variability but also the sea ice variability showed significant changes in winter, suggesting a possible linkage between the two (Yang and Yuan, 2014).

Overland et al. (2012) regarded AD as a strong driver of atmospheric circulation in early summer, and suggested that the Arctic sea ice has decreased by the series of negative AD years persistent during 2007-2012, although their study was limited in the analysis period that confined for the recent six years when the AD was in extremely negative phase. In extending this study, Serreze et al. (2016) examined the decadal changes in the SLP patterns, and the SLP anomalies in the recent years resembled more the negative AD pattern to which the sea ice decrease was attributed. However, there are not many pieces of research

based on the quantitative assessment of the relationship between SIE and the AD variability. This may be partly because the correlation between SIE and the AD index vanishes when the entire analysis period was applied since the 1980s.

This study focuses on the decadal change in the leading SLP modes in the Arctic. It aims to address the statistical relationship between large-scale atmospheric circulation and the Arctic sea ice variability in the interannual timescale and their decadal changes during boreal summer. Especially, the changes in atmospheric circulations in the mid-1990s and their impacts on the

Arctic sea ice are presented.

Section 2 presents the description of data and the analysis methods. Atmospheric circulation changes and their relationship with SIE are examined in Section 3. Section 4 provides further discussion on why the atmospheric circulation is highly related with the sea ice variability after the mid-1990s. Summary and conclusions are given in Section 5.

## 2 Data and Methodology

In this section, the data and methodology used in this study are described. In Section 2.1, the description of data for atmospheric variables and sea ice variables is shown, and then in Section 2.2, we introduce the methodology for analysis.

### 2.1 Data

5    SLP and wind data used in this study were obtained from the NASA Modern-Era Retrospective Analysis for Research and Applications, Version 2 (MERRA-2) atmospheric reanalysis (Gelaro et al. 2017) available from 1980 to present. MERRA-2 has a 1/2° (lat.) and 2/3° (lon.) horizontal resolution, with 72 vertical levels topped at 0.01 hPa. Surface wind reanalysis is defined at 2 m above the surface.

The National Snow and Ice Data Center (NSIDC) provides the monthly-mean SIE data as Sea Ice Index, Version 3 (Fetterer 10  et al. 2017). SIE is defined as the total area covered with at least 15 % concentration of sea ice for the given area. NSIDC also provides the sea ice motion (SIM) data (Polar Pathfinder Daily 25 km EASE-Grid Sea Ice Motion Vectors version 4; Tschudi et al., 2019) and sea ice age (SIA) data (EASE-Grid Sea Ice Age version 3; Tschudi et al., 2016). These data were produced from various sources of satellites; the Advanced Microwave Scanning Radiometer for Earth Observing System (AMSR-E), Advanced Very High Resolution Radiometer (AVHRR), Scanning Multi-channel Microwave Radiometer (SMMR), Special 15  Sensor Microwave/Imager (SSM/I), Special Sensor Microwave Imager Sounder (SSMIS), and National Centers for Environmental Prediction / National Center for Atmospheric Research (NCEP/NCAR) and the Buoy observations from the International Arctic Buoy Programme (IABP). The horizontal resolution of SIM and SIA is 25 km and 12.5 km, respectively, and the data cover the area from 48.4 - 90° N with Equal Area Scalable Earth Grid (EASE-Grid) for the period of 1978 to 2016 and 1984 to 2016, respectively.

20  Sea ice concentration (SIC) data were obtained from National Oceanic and Atmospheric Administration (NOAA) Optimum Interpolation Sea Surface Temperature version 2 (OISST v2; Reynolds et al., 2002), which has a horizontal resolution of 1° with the data period of 1982 – 2017.

This study also used the sea ice thickness data from the Pan-Arctic Ice Ocean Modelling and Assimilation System (PIOMAS) reanalysis produced by the Polar Science Center (Zhang and Rothrock, 2003). The original data with a horizontal resolution 25  of approximately 22 km were regridded onto the 1° Lat. x 1° Lon. grids for the analysis.

### 2.2 Methodology

The EOF analysis was conducted to obtain leading modes of atmospheric variability in SLP over the Arctic region (70°N-90°N) during summer (June – August), which is same methods to define the AD (Wu et al., 2006; Watanabe et al., 2006; Wang et al., 2009; Overland and Wang (2010). The analysis was first conducted for the entire analysis period of 1980-2017, and then 30  for the two separate periods of 1982-1997 and 1998-2017 to examine the decadal changes in the leading modes of atmospheric circulation. The separation of the analysis period before and after 1998 is identical to Yang and Yuan (2014). There was a

robust decadal change in the relationship between the atmospheric circulation and the sea ice extent in the mid-1990s. The leading three modes were analyzed in this study, which accounts for more than 80 % of the total Arctic SLP variability in summer. Due to the data availability, sea ice motion was analyzed just for 1982-201 6 and sea ice age was analyzed for 1984-2016.

The convergence of sea ice motion is calculated as,

$$-\nabla \cdot V = -\left(\frac{\partial}{\partial x}u + \frac{\partial}{\partial y}v\right),$$

where V is ice motion vector and each $u$ and $v$ indicate the zonal and meridional direction, respectively.

## 3 Result

In this section, the atmospheric circulation in the Arctic is introduced and then we describe the relationship between the atmospheric circulations and the sea ice with possible mechanisms.

### 3.1 Atmospheric Circulation in the Arctic

Figure 1 shows the interannual variation of the Arctic (70°–90°N) surface temperature (Fig. 1a) and SLP (Fig. 1b) anomalies during JJA, and SIE anomalies in September (Fig. 1c). Under global warming, there is an increasing trend in surface
temperature since 1998, which is about three times larger (0.16 °C decade$^{-1}$) than that for the period before 1997 (0.6 °C decade$^{-1}$). A more rapid decrease in SIE is also observed in the recent period (– 1.1 x 106 km$^2$ decade$^{-1}$), compared with the earlier period (– 0.4 x 106 km$^2$ decade$^{-1}$). This accelerated melting trend is reported by Comiso et al. (2008). On the other hand, the negative trend of SLP anomalies before 1997 turns into weakly positive after 1998.

We investigate if these changes in the trend of surface temperature, SLP, and SIE were caused by any changes in the large-
scale atmospheric circulation pattern in the mid-1990s. Figure 2a presents the climatological-mean surface wind pattern during JJA (vector) and the interannual variability of sea ice in September (shaded) as measured by the standard deviation of sea ice extent for the period 1982–2017. The secular negative trend in sea ice concentration was removed from each grid to consider only the interannual variation in 1982-2017. In summer, most of the sea ice variability is along the edge of the climatological-mean sea ice distribution (black line, denoted as the border of 15 % of sea ice concentration), particularly large in the Pacific
section such as in the Beaufort Sea and the East Siberian Sea. The time-mean surface wind shows the cyclonic (counter-clockwise) circumpolar circulation around the North Pole with an anticyclonic (clockwise) circulation over the Beaufort Sea. Figure 2b shows the difference in the interannual variability between the early and the recent period. Because of more sea ice loss in the recent period, the boundary of sea ice extent shifts poleward. This makes the region of interannual sea ice variability is moved northward and closer to the North Pole. The difference map of surface wind between the two periods shows a basin-
wide anticyclonic motion with a strong southward outflow to the North Atlantic. This surface wind pattern is consistent well

with the finding by Ogi and Rigor (2013), who attributed the rapid decreasing trend of SIE after 1996 to the anticyclonic surface wind pattern in the Arctic. Zhang et al. (2016) also suggested the intensification and stabilization of the Beaufort Gyre in the recent period. Comparing the boundary of sea ice area between the two periods, sea ice loss in the recent period is much larger in the Pacific section than in the Atlantic. As the sea ice area retreats poleward, it moves the center of action in sea ice variability to the north, showing a significant increase of the variability in the Beaufort Sea and in the East Siberian Sea. It is accompanied by sea ice variability decrease in the coastal sea in the Pacific section due to the overall retreat of sea ice. Noticeably, sea ice concentration over the Odden, a tongue of sea ice over the northeastern coast of Greenland (Shuchman et al. 1998), does not show a significant difference between the two periods. This spatial difference in sea ice melting is also shown in previous studies (e.g., Parkinson, 2014a, Cavalieri and Parkinson, 2012). It is intriguing what causes this asymmetric sea ice melting pattern, and this study further examines the change in the atmospheric circulation with a particular focus on the impact of the TDS on sea ice variation.

To obtain dominant variability modes of atmospheric circulation, the EOF analysis was conducted for seasonal-mean SLP anomalies in the Arctic. Figure 3 shows the three leading EOF modes. The first mode (EOF1) represents the AO, explaining ~ 50 % of the total variance. The corresponding principal component timeseries (PC1; Fig. 3d) is highly correlated with the AO index produced by the NOAA Climate Prediction Center (CPC) (r = 0.95). The center of the negative SLP anomalies is located over the Arctic and this anomaly extends to Greenland. The surface wind vector regressed onto the PC1 timeseries (Fig. 3a) shows a cyclonic circulation over the Arctic, corresponding to the positive phase of AO, with a large-scale southerly flow from the Atlantic, via Fram Strait, to the Arctic Ocean. The second mode (EOF2) shows a dipole structure, with one center over the Laptev Sea and the Kara Sea, and the other over the Beaufort Sea. Variation of this dipole pattern could enhance or reduce meridional flow crossing the Arctic and affects the TDS. This low-level flow may play a significant role in sea ice motion, as to be discussed later. The third mode (EOF3), which shows a comparable magnitude in variance comparing with that of the AD (17.2% vs. 11.5%), has not been much addressed in previous studies. This study defines this mode as A3, which is characterized by a cyclonic circulation over the East Siberian Sea and an anticyclonic circulation occupying over a broad area over from the Barents Sea to Greenland. This dipole SLP pattern is associated with the atmospheric flow squared to the TDS as it penetrates through the North Pole. The remaining modes from the EOF analysis account for much small variance lower than 6 %. Hereafter, this study focuses on the three leading modes (AO, AD, and A3) that explain almost 80 % of the total SLP variability, and the normalized PC timeseries are used for representing a temporal variation of each mode and the subsequent regression analysis. The positive phase is more dominant prior to 2007 in the AD index (PC2), and then it exhibits negative values successively, including the years of sea ice minima in 2007 and 2012. This feature is also supported by Overland et al. (2012). On the contrary, the A3 index does not seem to have a noticeable long-term trend.

## 3.2 Decadal Change in Atmospheric Circulation and Sea Ice

This study further examines if the atmospheric circulation pattern has changed significantly on a decadal scale, and its effects on the sea ice distribution change. The dominant atmospheric circulation patterns in the Arctic were again obtained from the

EOF analysis, applied for the two different periods, before and after 1998. It should be noted that the results were not sensitive when any year between 1996 and 2000 was chosen for the period separation. Figure 4 compares the leading EOF modes of monthly SLP anomalies for the two periods. The AO mode is captured as the first mode for both periods, followed by AD and A3, as in the sequence from the analysis to the entire period. Each corresponding PC timeseries shown in Figs. 5a–c (dashed

lines) has a high temporal correlation with those in Fig. 3, suggesting that each EOF modes are robustly identified as internal modes, regardless of analysis time. The significant difference is in the EOF spatial pattern of the AD. In the early period, the dipoles are located one in the Kara and the Laptev Sea and the other in the Beaufort Sea so that the surface wind across the Arctic is parallel to the dateline. On the other hand, the center of variability tends to shift eastward in the recent period. In particular, the variability maximum in the western hemisphere shifted from Queen Elizabeth Islands to Greenland. The pattern

correlation over the western hemisphere (60°-90°N, 0°-180°W) is estimated to test the statistical difference between the two patterns. Estimating the statistical significance with the effective sample size according to Bretherton et al. (1999; See supplementary information), the pattern correlation between the early and recent AD shows 0.58, which suggests the early and the recent AD patterns are not statistically same. On the other hand, the pattern correlation between the early and the recent AO patterns is around 0.99, suggesting no statistical difference. This change in SLP pattern accordingly changes surface wind

direction. Especially, it shows a clear development of the southerly flow in the recent period, blowing from the North Atlantic to the Arctic Sea via the Greenland Sea. This southerly flow does not exist evidently in the early period.

The change in the spatial pattern of the AD can impact on the sea ice variability significantly. As shown in Fig. 5d, the correlation between the AD and the September SIE timeseries dramatically increases in the recent period (r = -0.05 in the early period vs. r = 0.57 in the recent). Although we remove the linear trend in each PC timeseries and SIE before the correlation

analysis to avoid any spurious influenced by the trend, the result is not qualitatively different by including trend. It is also noted that, when we eliminate the SIE trend in each period for considering the rapidly declining trend in the recent, the correlation between the AD and the September SIE becomes much higher in the recent period (r = 0.57). This feature suggests that the SIE variability becomes more connected with the AD in the recent period. The strong relationship between the AD in boreal summer and the September SIE is consistent with the results from Wang et al. (2009). It is also noted that the correlation

between AO and SIE shows a statistically significant correlation at the 95 % confidence level for the entire analysis period (r = 0.33). Rigor et al. (2002) showed the AO exhibited a strong relationship with the Arctic sea ice in winter and Park et al. (2018) further highlighted the connection between wintertime AO circulation anomalies on the following summer sea ice extent. Williams et al. (2016) also suggested a lagged impact of the AO in winter affecting sea ice concentration in summer. This study hypothesizes that, although both the AO and the AD contribute to the interannual variability of the September SIE,

the strong relationship with SIE is largely due to the change in the spatial pattern of AD in the recent period. Responsible physical and dynamical mechanisms will be discussed in more detail in the next subsection.

Although our analysis focuses on the interannual variability, the result provides an implication on the recent negative SIE trend as well. The positive trend of SLP in the recent period (Fig. 1b) corresponds with the negative trend both in AO and AD (Fig. 3d and 3e, respectively). The decline of SIE caused by a more frequent occurrence of the negative AO seems to be related to

the warming and positive SLP anomalies in the pan-Arctic. This decline of SIE can be accelerated by a more frequent occurrence of the negative AD through enhancing TDS, which is discussed more in the next section.

We also note that the relationship of the sea ice with A3 is negligibly small for 1980 – 2017, where the correlation tends to decrease rapidly in the recent period. We focus more on the AD to account for sea ice variability in the next section.

## 3.3 Mechanisms of Sea Ice Response

We next analyze the September sea ice concentration (SIC) to explain the spatial distribution of sea ice variability. Figure 6 shows the regression pattern of September SIC onto the AD index (PC2) in each period. The regressed patterns are plotted with respect to the negative AD phase in order to better represent the southerly wind-induced ice loss over the Pacific sector of the Arctic. During the negative phase, the September SIC seems to increase over the Fram Strait in the early period with a

decrease in the Pacific sector over the East Siberian Sea, the Chukchi Sea, and the Beaufort Sea in the recent period. Comparing SIC between the early and the recent period, sea ice over the Fram Strait remains almost the same in September, suggesting less contribution to the entire Arctic SIE. In contrast, the SIC reduction in the Pacific sector is more dominantly contributing to the decrease of entire Arctic SIE.

Atmospheric circulation associated with the AD is closely linked to the 850-hPa temperature change (See Supplementary Fig.

S1). As the center of SLP anomalies shifts eastward in the recent period, the warm area over the Beaufort Sea moves to Greenland. Considering the surface wind anomalies in Fig. 6, the Greenland region seems to be affected by warm temperature advection from the southerlies along the Baffin Bay (See Supplementary Fig. S2). However, these pattern changes cannot explain increased sea ice melting over the Pacific sector of the Arctic in the recent period as shown in Fig. 6. Therefore, this study further examined the decadal changes in the net downward surface heat flux associated with the AD. The surface heat

flux change driven by AD (Fig. 7) corresponds well with the sea ice concentration change pattern shown in Fig. 6. The net downward surface heat flux anomalies are negative in the sea ice increasing region such as in the Atlantic sector of the Arctic Sea, and positive for the rest of the regions where the sea ice decreases. The surface heat flux anomalies are mostly contributed by the changes in the shortwave radiation terms (See Supplementary Fig. S3). In particular, the upward shortwave radiation tends to decrease significantly in the East Siberia, Chukchi, and Beaufort Seas where the sea ice melting signal by AD is

pronounced in the recent period (Fig. 6b). This process is also reflected in the reduction of surface albedo due to the sea ice decrease (Fig. 8). The response of surface albedo becomes larger and much clearer in the recent period while the spatial pattern of surface albedo regression onto the AD index is essentially the same. Although the response in the surface heat flux suggests the positive sea ice-albedo feedback process engaged in the Arctic sea ice variability associated with the AD, it remains the question what mechanisms are primarily responsible for triggering this feedback process.

The difference in the SIC response in the Pacific sector seems to be related to the change in sea ice dynamics. In summer, sea ice tends to move along the geostrophic wind (Thorndike and Colony 1982; Ogi et al., 2008). The sea ice motion associated with the AD (c.f. Fig. 9a and 9b) becomes faster in the central Arctic, particularly around the boundary of the sea ice extent (Fig. 9c) in the recent period. Through this, the sea ice is drifted more toward the Norwegian Sea and discharged to the North

Atlantic. This sea ice motion change is consistent well with the change in the surface wind driven by the AD (c.f. Fig. 9d and 9e), suggesting that the sea ice is drifted toward the direction of the surface wind to a good approximation. Although the sea ice motion can be depicted only over the sea ice-covered area, the sea ice motion can be inferred from the pattern of surface wind in their final stages in discharging and melting in the North Atlantic. This feature is also evident in the sea ice motion divergence patterns, in which the divergence anomalies becomes larger in the Chukchi and the Beaufort Seas due to more enhanced transpolar sea ice drift motion toward the Atlantic sector (Figure 10).

Comparing the surface wind pattern associated with the AD in the past and the recent periods (Fig. 9f), the largest difference is shown over the Fram Strait. In the past, the transpolar wind is backing in the Atlantic side of the Arctic Sea (i.e., Barents and the Kara Sea, Fig. 9d), whereas it extends further south to the Atlantic in the recent (Fig. 9e). This enhanced outflow should be responsible for the sea ice drift and large sea ice discharge through the Fram Strait (Fig. 9b). From these figures, northerly winds have been strengthened from the central Arctic to the North Atlantic in the recent period to provide a more favorable condition for sea ice to be discharged to the North Atlantic.

Another contribution may come from the change in oceanic heat transport. The surface wind over the Chukchi Sea is meridional in the past period, while it becomes zonal in the recent period (Figs. 9d and f). The sea ice over the Pacific sector may be affected by relatively warm water inflow from the North Pacific through the Bering Strait (Shimada et al., 2006; Woodgate et al., 2010; Carmack et al., 2015; Serreze et al., 2016b), and this process can be enhanced in the recent period by increased zonal component of surface wind blowing over the Chukchi Sea. The zonal surface wind anomalies can provide a more favorable condition for the warm water transport in a meridional direction toward the Arctic sea ice regions via Ekman transport. The enhanced zonal wind component in the Pacific sector is connected with the intensification of the anticyclonic circulation anomalies over the Beaufort Sea as shown in Fig. 6. Wu et al. (2014) also indicated a significant intensification of the Beaufort High during summer and autumn and attributed it to the rapid decline of sea ice over this region in the recent decade. This mechanism seems to work during 2007-2012 when the Beaufort High was persistently strong (Overland et al. 2010). It also seems to explain relatively weak sea ice melting during the recent three summers (2015-2017), when the cyclonic circulation anomalies were likely to prevent sea ice from melting.

One of the remaining factors to be considered is the changed property of Arctic sea ice. The sea ice becomes younger (Fig. 9c) and presumably thinner in thickness. This thickness change is also confirmed well by the PIOMAS reanalysis data (Fig. 9f). It suggests the Arctic sea ice becomes more vulnerable to the dynamical forcing such as atmospheric wind and oceanic heat transport, especially in the melting phase.

## 4. Further Discussion

Section 3 examined the relationship between atmospheric circulation and the sea ice variability in summer and suggested the possible mechanisms. Our result is in line with the previous studies that suggested the important role of atmospheric circulation impacting on sea ice (i.e. Rigor et al., 2002; Wang et al., 2009). This study further suggests that it is the AD pattern change since the mid-1990s that affects the significant change in September SIE in the recent period.

This study suggests two significant changes in the recent period (Fig. 9). Firstly, the Arctic sea ice becomes much vulnerable, especially in the Pacific sector as it is placed with younger and thinner sea ice. Secondly, the large-scale atmospheric circulation pattern has been changed. In summer, the sea ice motion is affected by the surface wind so that the change of surface wind can impact the change in sea ice motion.

Regarding the sea ice dynamics, the surface wind along the TDS might have a strong effect on sea ice outflow. The sea ice outflow modulated by the surface wind along the TDS is relatively well recognized by previous studies (i.e. Watanabe et al., 2006; Wang et al., 2009), which suggest that the AD is linked closely with the sea ice variability. From our analysis, however, the AD shows no significant relationship with the September SIE in the early period, although SIM is consistent with the direction of TDS. Our result highlights that the surface wind pattern linked with the AD has changed and this is suggested to

be a primary factor responsible for increasing sea ice variability in the recent period. In the Atlantic sector, the left-curved surface wind pattern in the south of the Fram Strait may result in very little sea ice outflow to the North Atlantic in the past period. On the other hand, the surface wind in the recent period turns into northerly stretching to the North Atlantic during the negative phase of AD, which all seem to be favorable to the increased disposal of sea ice into the warmer North Atlantic. The negative SIC anomalies in the Pacific sector seems to be related to the transport of sea ice along the TDS to the North Atlantic

during the negative AD phase.

The remaining question is what causes the spatial pattern change of AD, which is a key to enhance the variability of sea ice that becomes thinner and younger by global warming. Although many previous studies highlight the climate regime shift in the mid-1990s, the reason for the AD change is unclear. It is hypothesized that the phase change of the Pacific Decadal Oscillation (PDO; Mantua et al., 1997) could be responsible for it, which phase has changed from positive to negative between

1997 and 1998. It is the time coincident with the regime change in the Arctic in our study (c.f. Fig. 1). Figure 11 shows the PDO index is 1948 to 2017. The PDO index is obtained from the NOAA National Centers for Environmental Information (NCEI).

To examine the effect of PDO, we first separated the years depending on the PDO phase as in Table 1. The positive PDO years are defined when the PDO index is above 0.5 and negative below -0.5 from 1948 to 2017. For each sampled year, we conducted

the EOF analysis again to the monthly SLP anomalies to obtain the dominant variability depending on PDO. Figure 10 shows the two leading EOFs for the positive and the negative PDO phase, respectively. In this case, we used the NCEP/NCAR reanalysis 1 data due to its long-time availability (Kalnay et al., 1996). Regardless of the PDO phase, the two dominant modes of SLP variability are AO and AD robustly. It is noted that the AD pattern is different significantly according to the phase of PDO. When the PDO is in the negative phase, the one center of action moves to Greenland, which is consistent with our

findings in Fig. 4. For the negative PDO years, we also conducted the same EOF analysis without recent years (1999-2017) and found a qualitatively similar result, with the center of action shifted to Greenland (See Supplementary Fig. S4). Dynamical mechanisms how the negative PDO shifts the AD center is still unclear from our analysis. Nevertheless, the statistical relationship between PDO and AO is quite robust and depends less on the data analysis period, suggesting a possible role of PDO in modulating the AD pattern.

## 5 Summary and Conclusion

How the large-scale atmospheric internal variability could affect the Arctic sea ice variability has been documented relatively well by previous studies. For example, the winter AO can modulate the summer sea ice variability (Rigor et al., 2002; Park et al., 2018) and the strong negative AD is considered as the major driver for sea ice minimum in 2007 summer (Wang et al., 2009). This study examined the relationship between the sea ice and the Arctic atmospheric circulations separated by EOF analysis. From the EOF analysis of monthly SLP anomalies in summer, three leading patterns are extracted, the AO as the most dominant one, the AD as the second, and the third mode that is not discussed much in the previous literature (defined as A3 in this study). The AO and the AD have a significant relationship with the September SIE for 38 years (1980-2017) as a consistent result with Rigor et al. (2002) and Wang et al. (2009).

To explore if there are any significant decadal changes in the dominant atmospheric variability in the Arctic, we separated the analysis for the early period (1982-1997) and the recent period (1998-2017), respectively. In both periods, the three atmospheric circulations (AO, AD, and A3) are robustly separated by the same EOF analysis. The noticeable change between the two periods is that the AD pattern is shifted in eastward in the recent period. The center of the action on the Beaufort Sea moves to Greenland. This change in the AD pattern is responsible for the sea ice variability. The correlation coefficient for September SIE is dramatically increasing in the recent period as 0.57, whereas the AD is not much related to the September SIE in the early period. At the interannual scale, AO and A3 do not show a significant relationship with the September SIE in both periods.

In the recent period, the surface wind on the Beaufort Sea associated with the AD seems to be favorable to drive warm Pacific summer water to the sea ice edge. In addition, due to global warming, the sea ice becomes thinner and younger, to be more vulnerable to the large-scale atmospheric circulation. Thus, the anti-cyclonic circulation anomalies on the Beaufort Sea transports the warm Pacific water to sea ice edge by Ekman transport and increase sea ice melting in the Pacific sector. In addition, the strengthened TDS during the negative AD phase enhances the meridional transport of melted sea ice across the Arctic and outflows to the North Atlantic Ocean.

The sea ice concentration change associated with AD is also accompanied by the change in the net surface heat flux, suggesting an important role of thermodynamical process as well. The surface heat flux change is mostly contributed by the changes in the shortwave radiation and surface albedo in the prominent sea ice melting regions, suggesting a positive sea ice-albedo feedback in the recent period associated with the AD.

This study implies that the relationship between atmospheric circulation and sea ice has a decadal change in the mid-1990s. Although AO and AD have a significant relationship with sea ice variability at the 95% confidence level during the entire period (1980-2017), only the AD in recent period shows a strong relationship with the Arctic sea ice. The reason why the AD is changed is a remaining question, however, we suggested the PDO is one of the possible modulators to the spatial pattern change of AD.

**Acknowledgments**

This work was funded by the Korea Meteorological Administration Research and Development Program under Grant KMI2017-02410

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

**Table. 1. Selected PDO years in this study.**

| PDO Phase | Years |
|---|---|
| Positive | 1957, 1958, 1981, 1983, 1987, 1992, 1993, 1997, 2015 |
| Negative | 1949, 1950, 1951, 1952, 1955, 1956, 1961, 1962, 1963, 1964, 1967, 1970, 1971, 1972, 1973, 1975, 1978, 1991, 1994, 1998, 1999, 2000, 2001, 2002, 2008, 2009, 2010, 2011, 2012, 2013 |

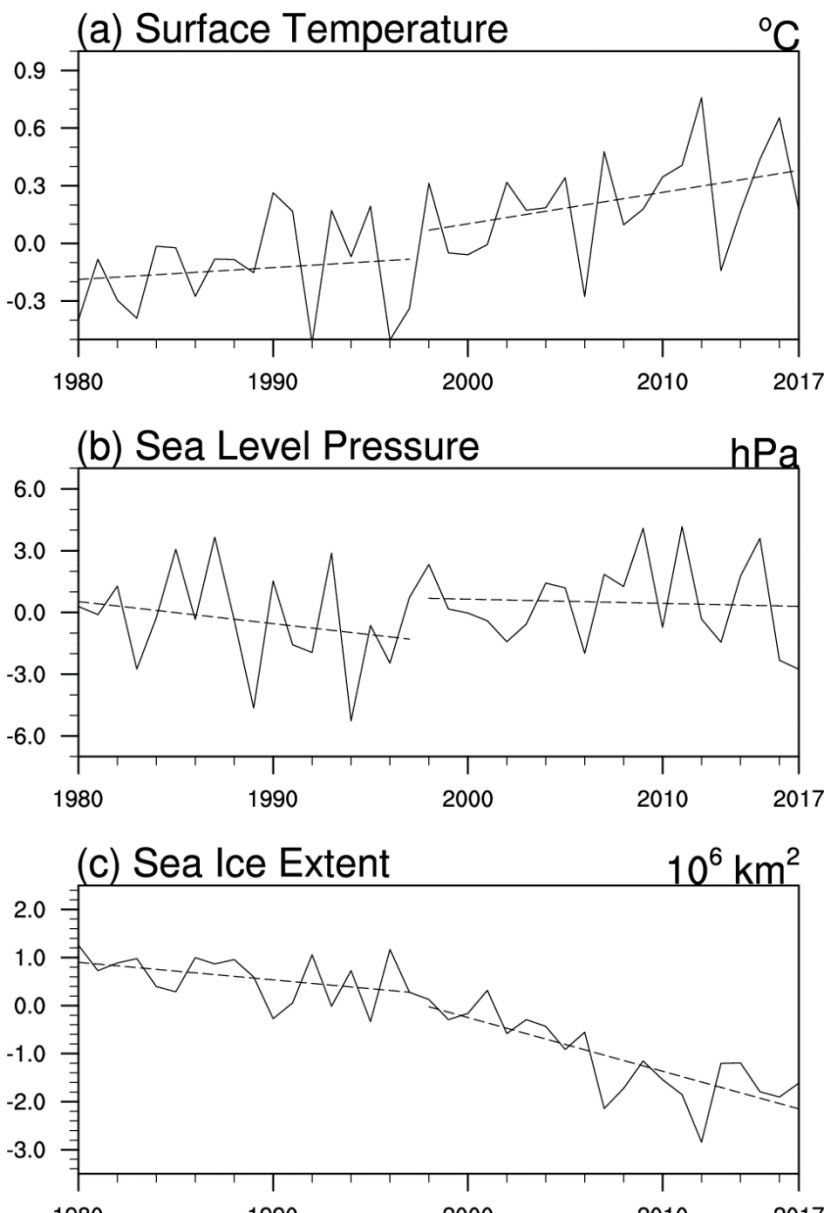

Figure 1: (a) Surface temperature and (b) sea level pressure anomalies in the Arctic region north of 70° N during June-July-August (JJA). (c) is the sea ice extent (SIE) in September in the same area. The anomalies are the departures from the average of 1981-2010. Dashed line shows the trend before and after 1998.

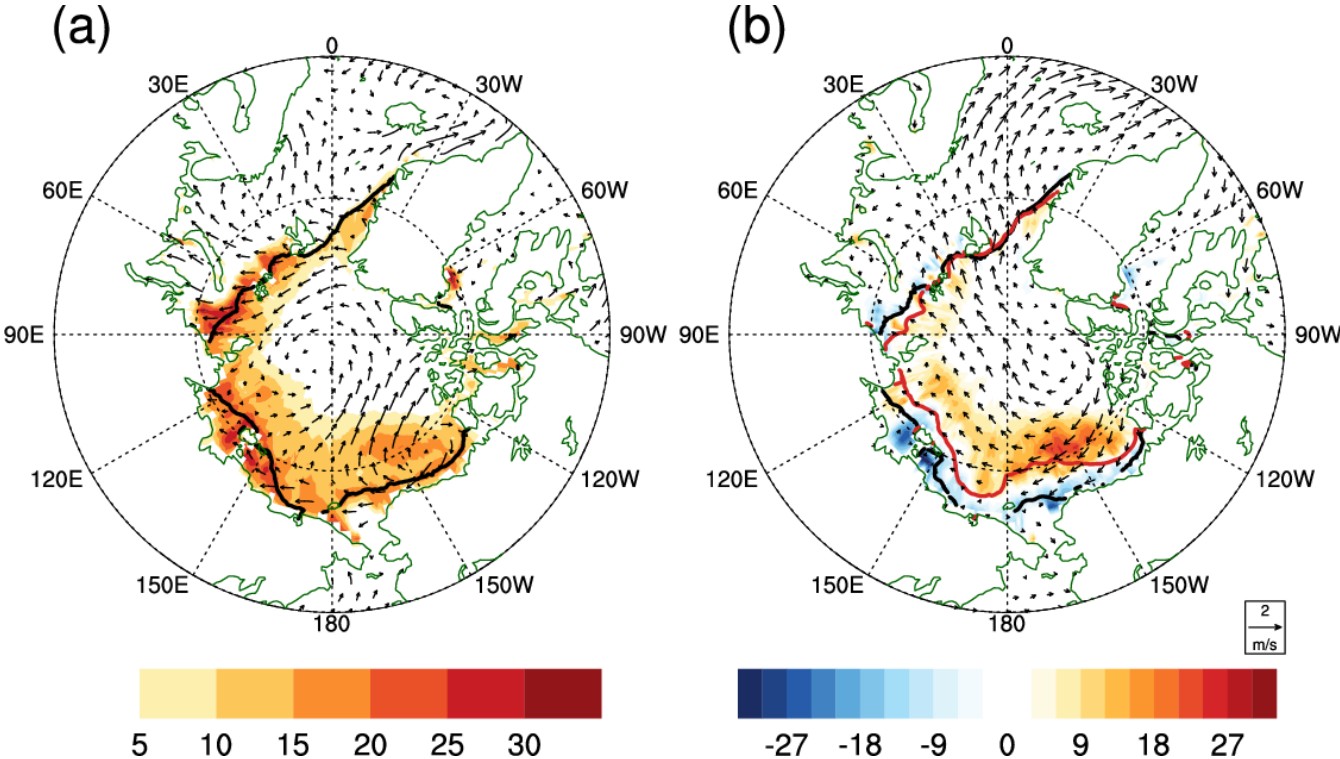

**Figure 2: (a)** Climatological-mean (1982-2017) pattern of surface wind at 10 m during JJA (vector, m s⁻¹) and the SIE in September (solid line). The shaded area shows the standard deviation from the detrended yearly variation of September SIE in each location. **(b)** The difference of time-mean surface wind (vector, m s⁻¹) and the standard deviation of September SIE between the recent and the early period (1998-2017 minus 1982-1997; shaded). The solid lines indicate the SIE for the early (black) and the recent (red) period, respectively.

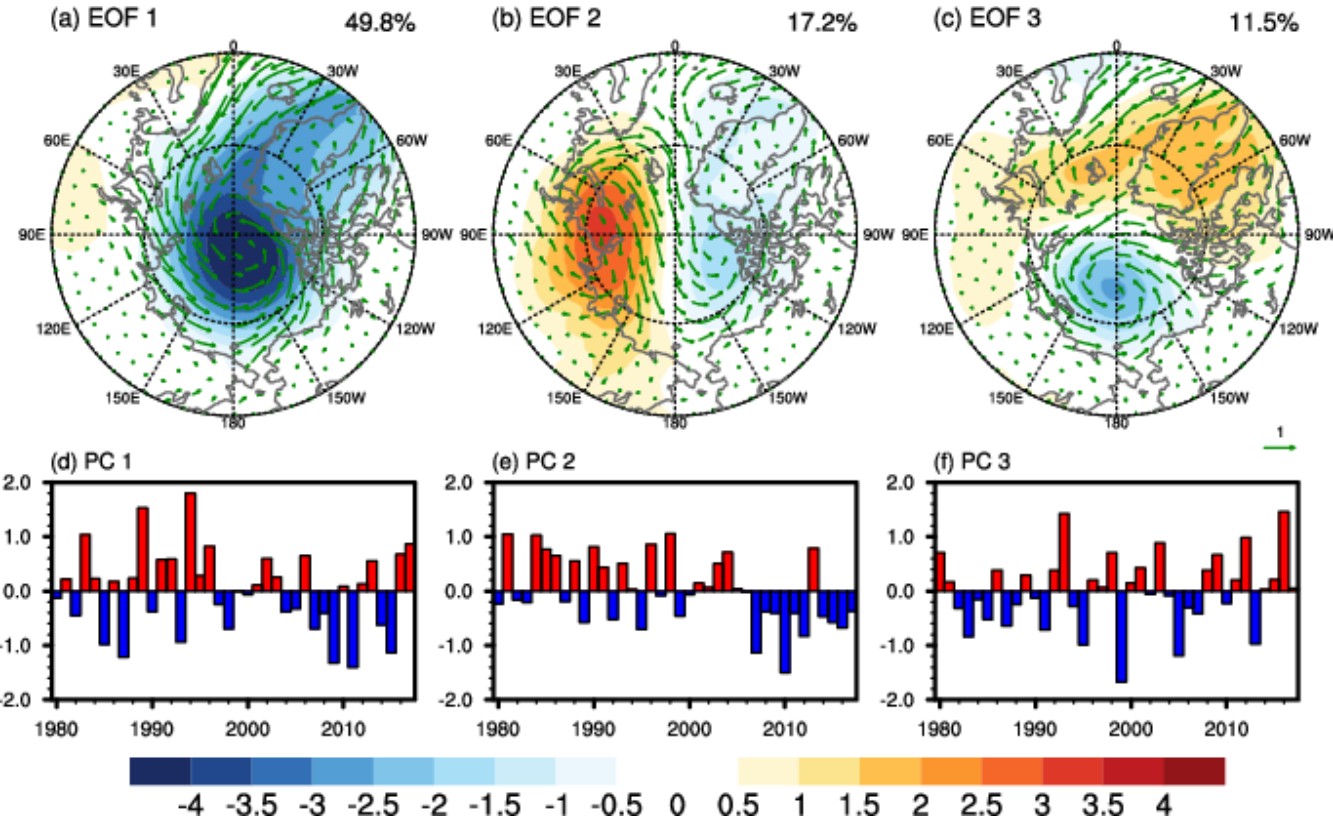

**Figure 3:** Leading EOFs of JJA-mean SLP anomalies (shaded) for (a) AO, (b) AD, and (c) A3. (d), (e), and (f) are the corresponding timeseries, respectively. The regression patterns of surface wind onto the corresponding EOF timeseries are also given in (a)-(c) (vector).

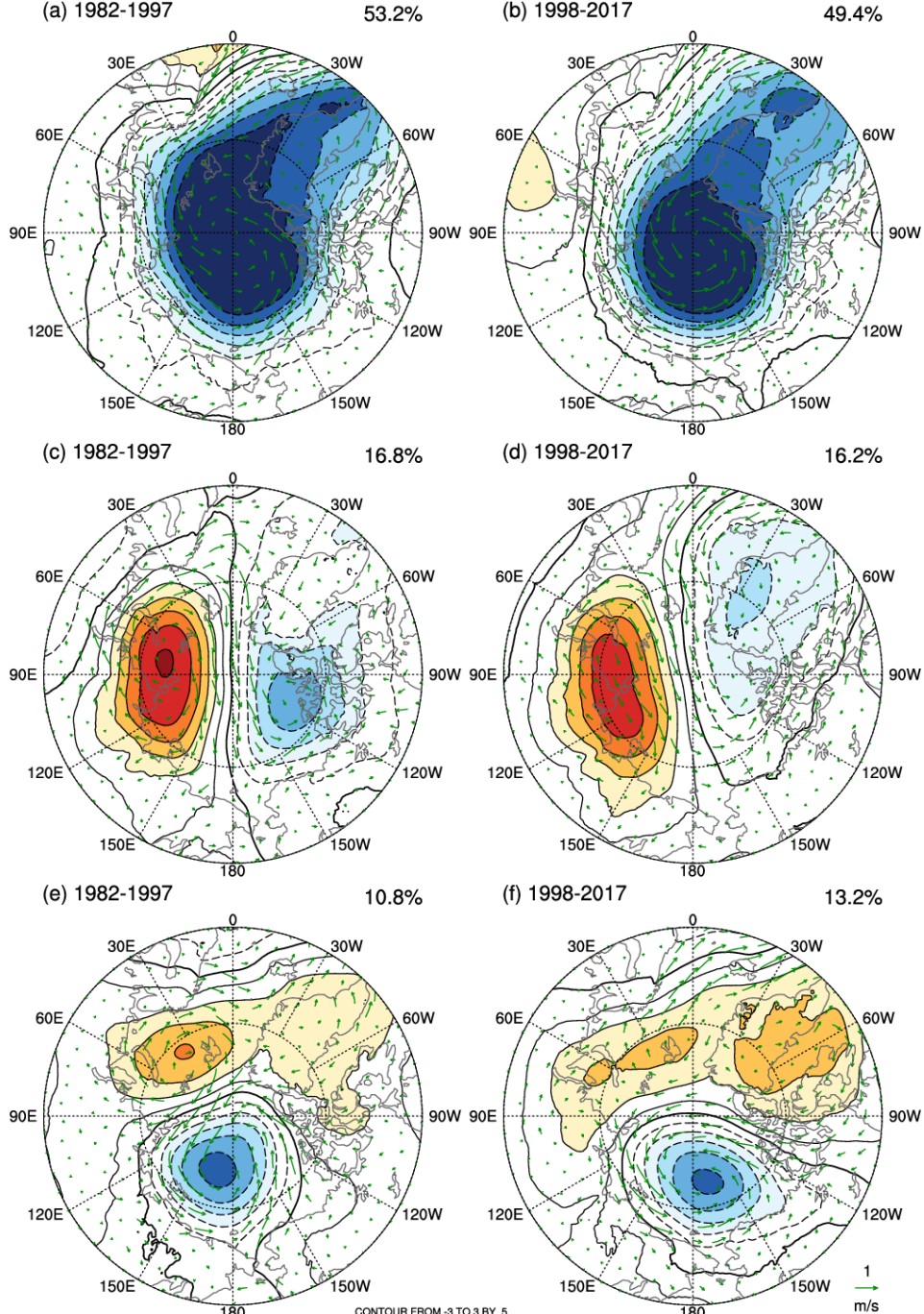

**Figure 4: The three leading EOFs of JJA-mean SLP (shaded) in the early (1982-1997, left panels) and the recent (1998-2017, right) period. (a) and (b) for the first mode, (c) and (d) for the second, and (e) and (f) the third mode, respectively. The regression pattern of surface wind anomalies (vector) is also shown in each map.**

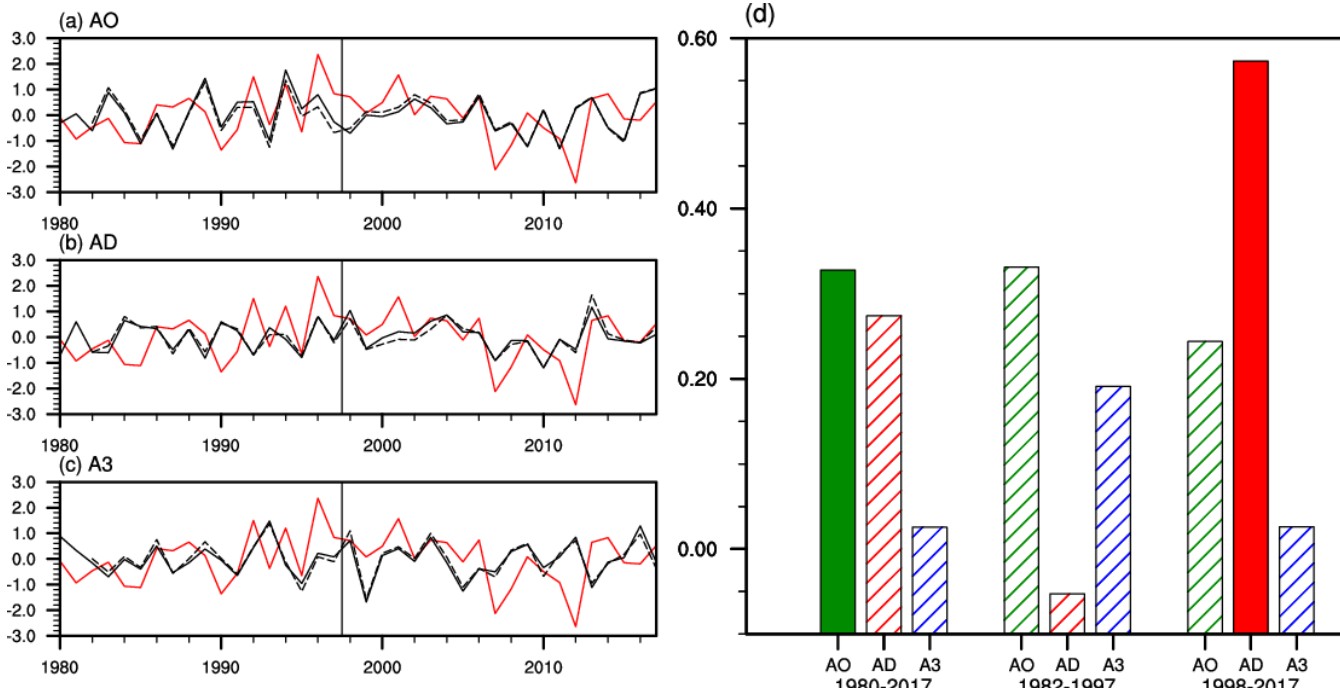

**Figure 5:** Timeseries of the three SLP EOFs (solid black) for (a) AO, (b) AD, and (c) A3. Timeseries of September SIE are also presented in each figure (solid red). In (a)-(c), the EOF timeseries from the separate analysis for the period before and after 1998 are also compared (dashed lines in black). (d) is the correlation coefficient between each EOF timeseries and the September SIE for the entire (1980-2017), early (1982-1997), and recent (1998-2017) period, respectively. A linear trend in each EOF timeseries was eliminated before calculating correlation. Filled color indicates statistically significant at the 95% confidence level.

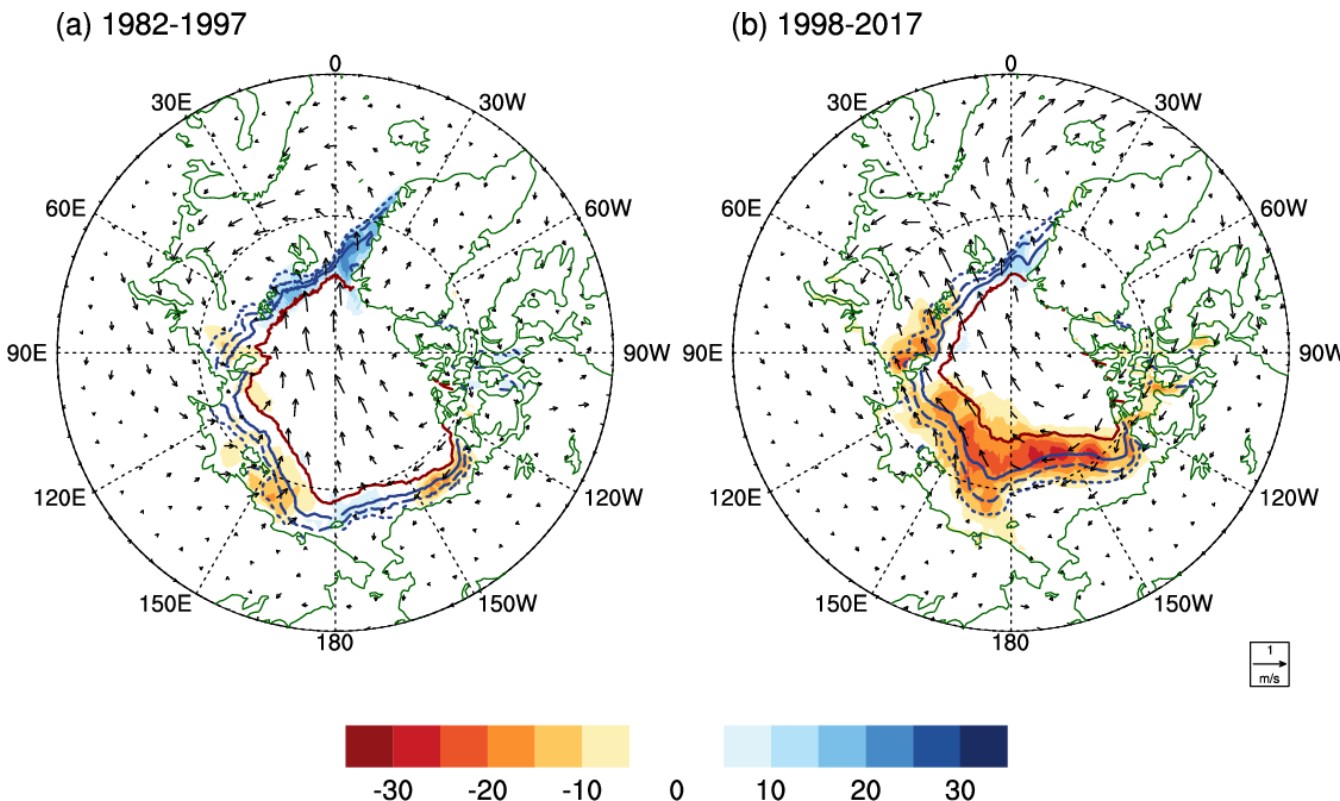

**Figure 6:** Regression patterns of sea ice concentration (shaded, %) and the surface wind anomalies (vector, m s$^{-1}$) onto the AD index for (a) the early (1982-1997) and (b) the recent (1998-2017) period. The AD index was reversed in sign for the melting phase of sea ice. Colored lines indicate the time-mean sea ice concentration of 15 % (red dotted), 30 % (red dashed), 50 % (red solid) and 80 % (blue solid) in each period.

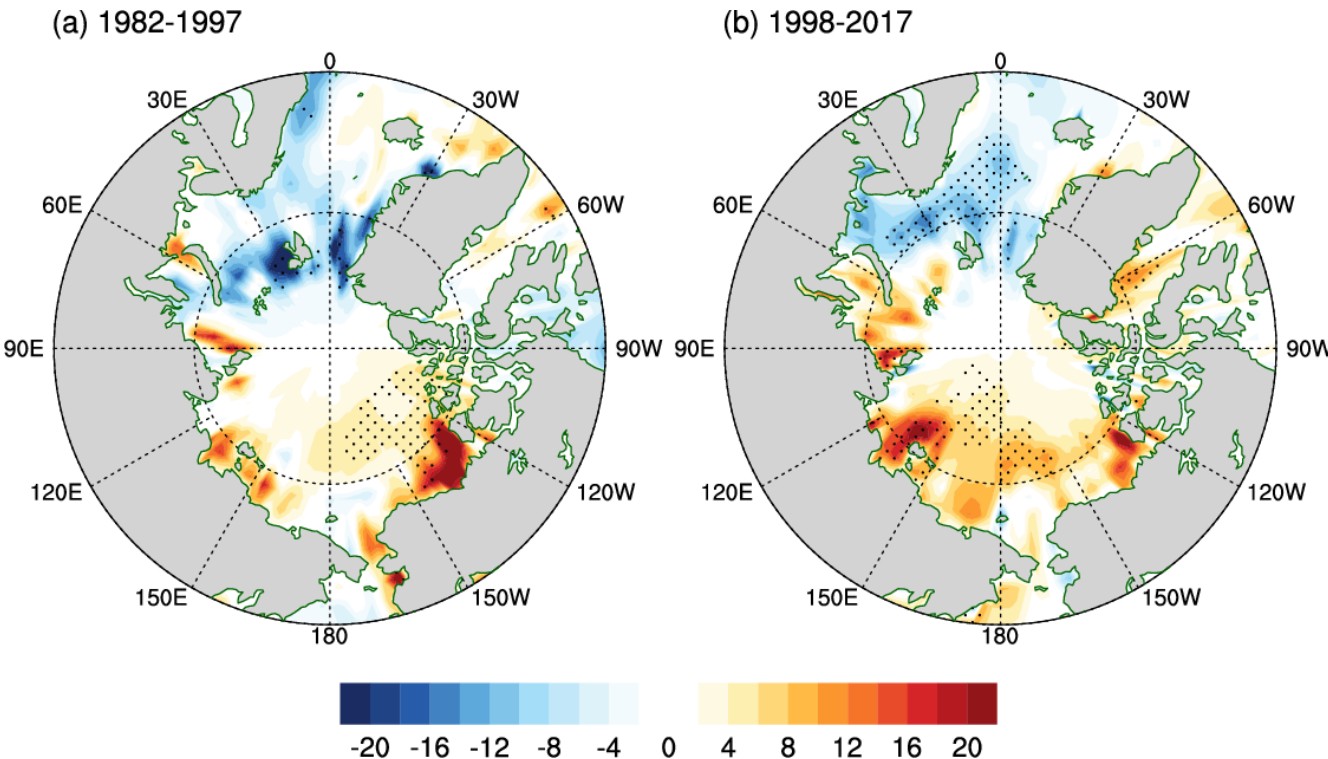

**Figure 7. Regression patterns of net surface heat flux (sensible + latent + net short wave + net long wave, W m$^{-2}$) onto the AD index for (a) the early (1982-1997) and (b) the recent (1998-2017) period. The heat flux is defined as positive for the downward, and the AD index is reversed in sign for representing the melting phase of sea ice. Dotted area indicates the statistically significant region at the 95% confidence level.**

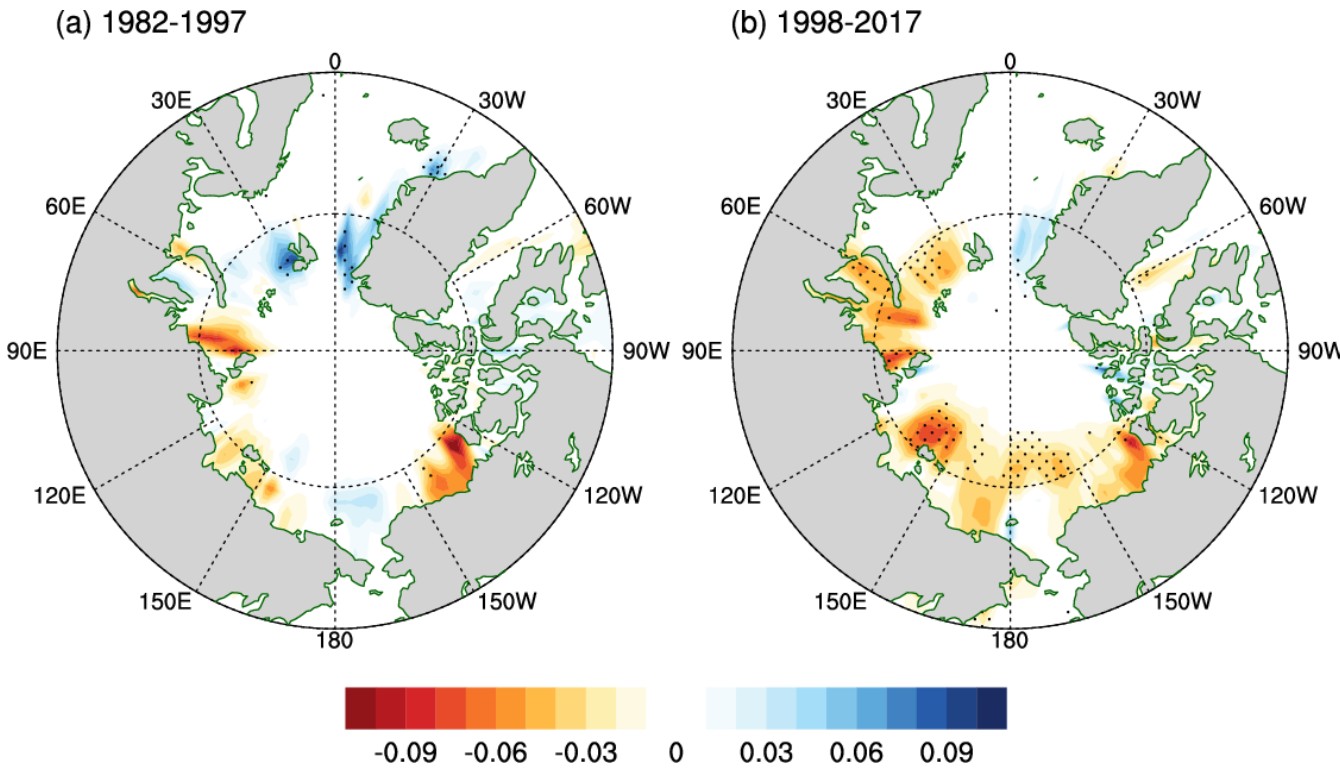

**Figure 8. Surface albedo changes associated with the summer AD in (a) the past (1982-1997) and (b) the recent (1997-2017) periods, respectively. The AD index is reversed in sign before regression.**

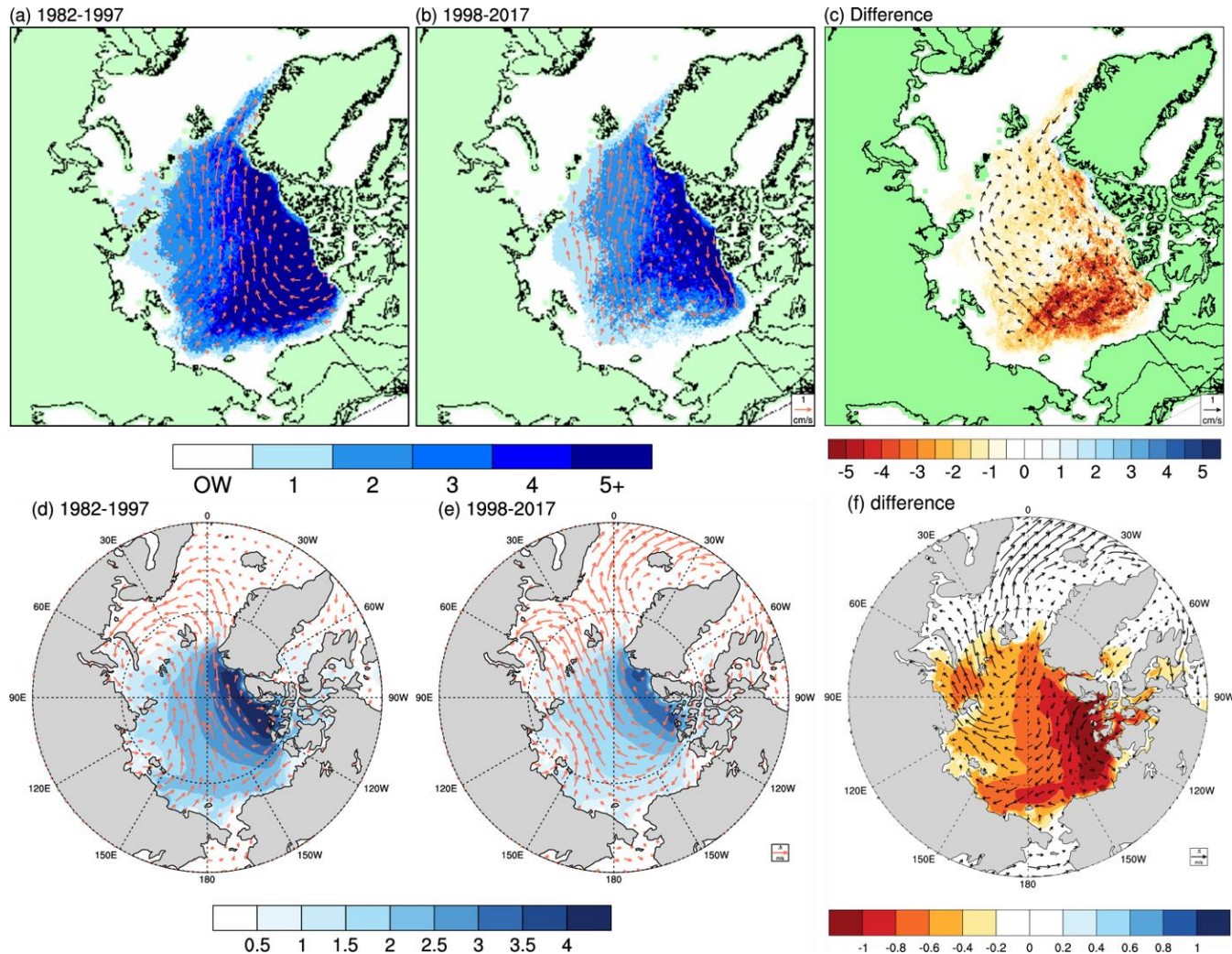

**Figure 9: Regression pattern of sea ice motion onto AD index and its difference (top) and regression pattern of surface wind onto the AD index and its difference (bottom) in the early (1982-1997, left) and the recent (1998-2017, center) period, respectively. Shaded is the sea ice age (top) and the sea ice thickness (bottom) in September averaged over each period and its difference.**

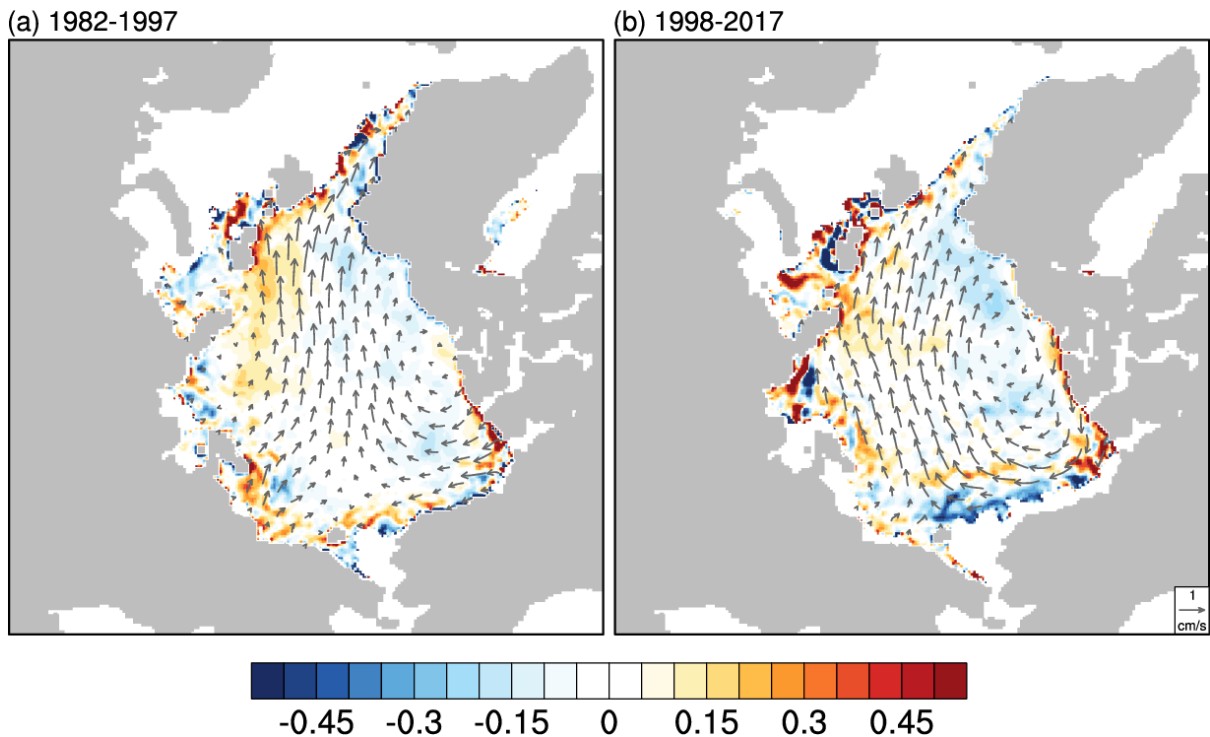

**Figure 10. Regressed sea ice motion vector (arrow, cm s$^{-1}$) and its convergence (shaded, 100 day-1) onto the AD index in (a) the early (1982-1997) and (b) the recent (1998-2017) period, respectively.**

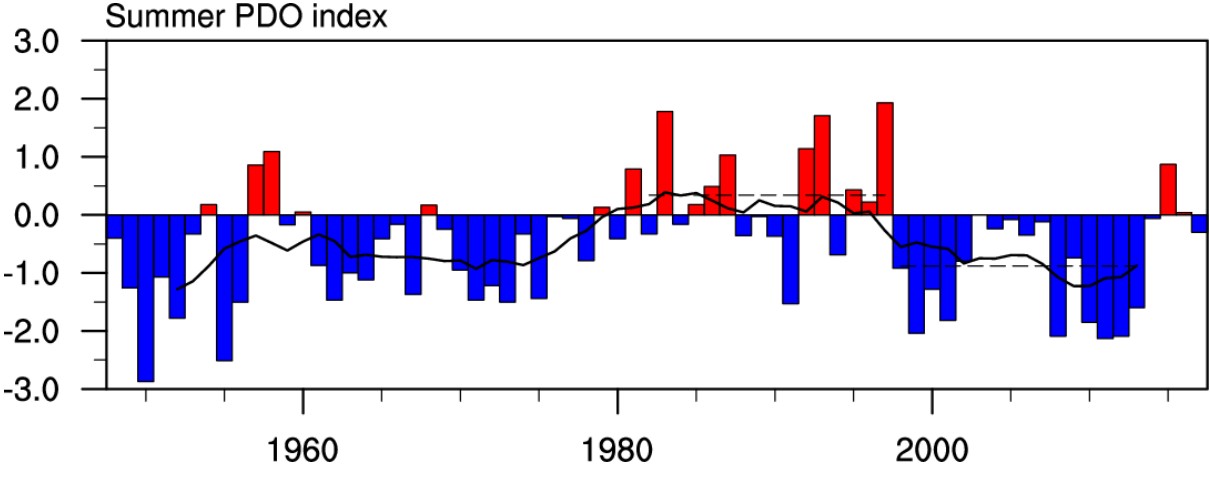

**Figure 11: The PDO index in summer (JJA) from NOAA National Centers for Environmental Information (NCEI). Solid line indicates 9-yr running average and the dashed line indicates the average over 1982-1997 and 1998-2013.**

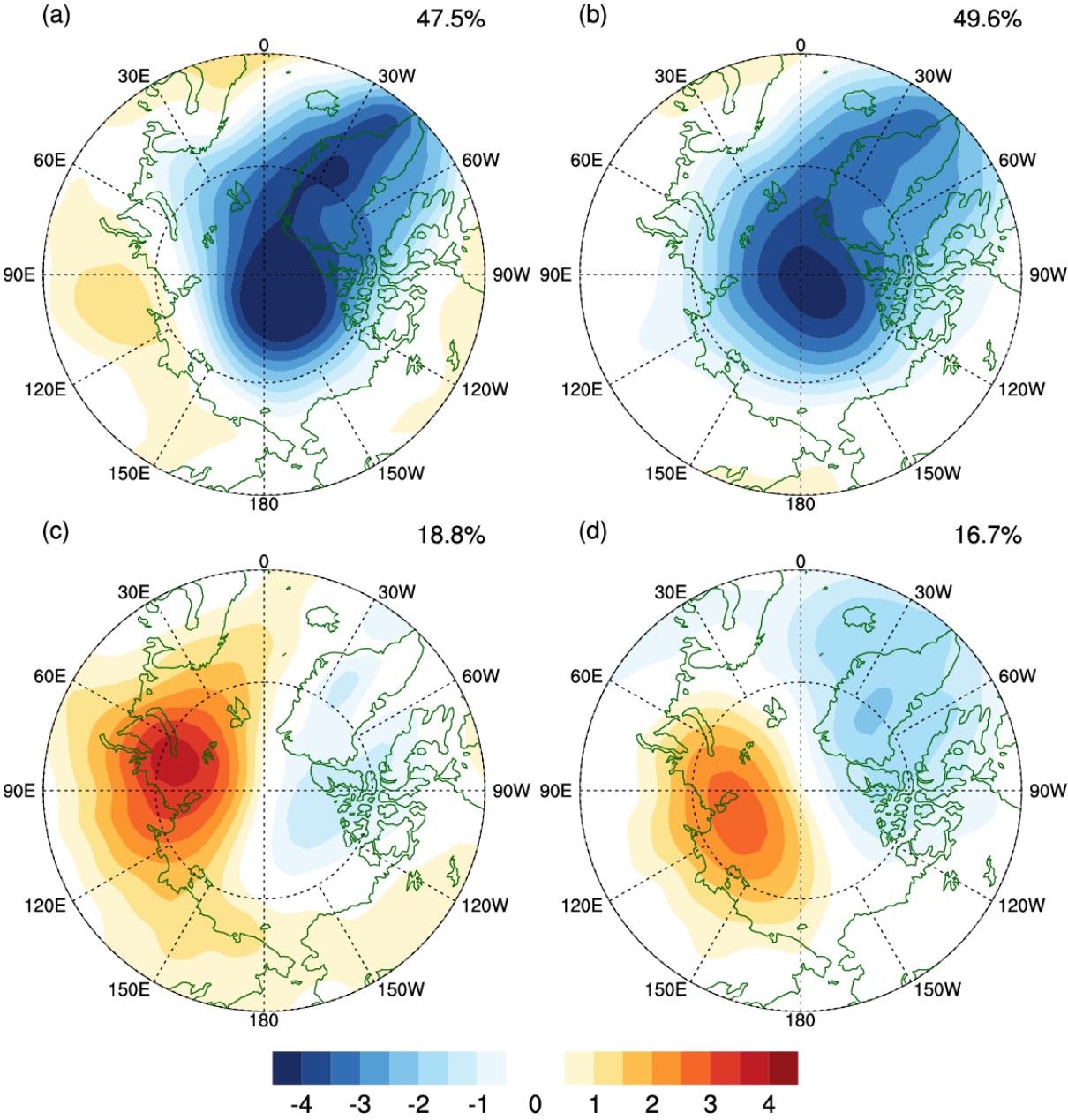

**Figure 12: Leading EOF modes according to the PDO phase. (a) and (c) are the EOF1 and EOF2 in the positive PDO years, and (b) and (d) are the EOF1 and EOF2 in the negative PDO years. The positive PDO years were defined when the PDO index is above 0.5 and negative below -0.5. The PDO index was obtained from NCEI. See the text for detail.**