# Peer review of "Decadal Changes in the Leading Patterns of Sea Level Pressure in the Arctic and Their Impacts on the Sea Ice Variability in Boreal Summer"

_The Cryosphere, 2019_

## Referee Comment (RC1) · Anonymous Referee #1 · 17 Apr 2019

General

This paper is telling is that September sea ice conditions are strongly shaped by atmospheric circulation patterns during summer, and that atmospheric circulation patterns are variable and have shifted over time. We have known this for many years, and a number of previous efforts have also noted that as the ice thins, relationships between atmospheric circulation anomalies and sea ice responses may be changing. In this sense, the present paper, while impressive in terms of depth of analysis, isn't really telling us anything fundamentally new.

Specific

[Figure]

Abstract, Page 1, Line 15: What month is this correlation based on? September ice extent against summer circulation? Be specific.

Page 1, line 24: The downward trends in sea ice extent involve more than "rapid melting".

Page 1, Line 25-28: Over what period did Serreze et al. [2007] compute the trend? The 12.4% per decade trend cited in later studies is not "expedited", it is simply based on a longer sea ice record. Also, percent per decade trends are meaningless numbers without clearly citing the baseline averaging period.

Page 1, Line 29: Surface warming is not Arctic amplification – AA refers to a comparison between temperature trends between the Arctic and the globe as a whole (or the northern hemisphere). And there seems to be a misunderstanding here – a large component of AA seems to be due to ice loss (the ocean loses heat to the atmosphere in autumn and winter), rather than the cause of it.

Page 2, line 3 and elsewhere in the text: A "declining trend" implies that the trend is getting smaller. The correct term is "downward trend"

Page 2, line 5: To state that the underlying mechanisms for sea ice variability in summer are still "under debate" is quite a stretch. Scientists have been looking at these mechanisms (atmospheric and oceanic variability) for many years. The authors should be citing earlier pioneering studies – from the way the text reads, there was no research on mechanisms behind sea ice variability before the dawn of the 21st century. We all stand on the shoulders of those before us. Give credit where credit is due.

Page 3, lines 1-3: As I recall, Ogi et al. [2007, 2008] were discussing what called the "summer AO" pattern, not the AD.

Page 4, line 12: I leave it to another reviewer to comment on the validity of conducting an EOF analysis over such a very restricted spatial domain (70 to 90 deg. N).

Page 4, line 25: Again, Arctic amplification is not about the Arctic temperature trends

alone, it's about the comparison between Arctic and global temperature trends. And it's not "polar amplification" - it's just the Arctic.

Page 5, lines 14-15. Assuming that the "Pacific section" refers to the Beaufort/Chukchi seas, why would there be a greater ice loss here in the later period when the motion is more onshore than in the previous period and would tend to transport thick ice from north of the Canadian Arctic Archipelago into the region?

Page 6, line 5: As far as I can see, no trend analysis has been performed on the time series. And I don't see much of anything resembling decadal scale variability in PC1. What I see is a series of ups and downs.

Page 6, lines 15-25: I have a very hard time convincing myself that the patterns for the earlier and later periods shown in Figure 4 are very different. In my opinion the authors are trying to read too much into these figures.

Page 7, line 16: Is melting the only thing going on here?

Page 8, line 1: The differences in ice motion between the two periods seems very nuanced to me. Again, I get the impression that that the authors are trying to read too much into the differences.

Page 8, line 12: Rebecca Woodgate has a number of papers addressing links between the Bering Strait heat inflow and sea ice conditions in the Chukchi Sea and potentially beyond. Also see: 10.1002/2016JC011977, which specifically examines predictability of ice conditions in the Chukchi Sea based on the Bering Strait heat inflow.

Page 8, line 15 It needs to be acknowledge here (or somewhere ) that the last three summers have been very cyclonic over the central Arctic Ocean; in other words, the much bandied "intensification" of the Beaufort Sea high appears to have broken down.

Page 9, lines 16-33: I think it is very difficult to argue that the cause of the shift in the AD is due to a phase change in the PDO. All that one can really say is that the shift in the AD (which seems minor to me) is part of a large-scale pattern of change involving

the PDO. The link is certainly interesting, but I'm hesitant to read too much into cause and effect.
* * *

---

## Referee Comment (RC2) · Anonymous Referee #2 · 29 Apr 2019

This study examines the rapidly increasing influence of summer Arctic dipole mode (AD) on September sea ice extent over the last decade using reanalysis and observational data. The authors show that the negative AD event has been more frequent since mid-2000's (Fig. 3), and has strongly influenced September sea ice extent (Fig. 5) by decreasing sea ice cover over the Pacific sector of the Arctic (Fig. 6). The authors further present that the increasing influence of AD on summer sea ice cover is partly because of sea ice thinning (Fig. 8), which increases the sensitivity of sea ice cover to southerly winds.

This study nicely expands on the work of Wang et al. (2009), but the main conclusions of this study are somewhat redundant with those of Serreze et al. (JGR 2016), which carefully analyzed the relationship between AD and summer Arctic sea ice extent. While it is a little difficult to argue that this study has enough novelty to justify publication at this stage, I think this study has great potential to become an influential paper. I am optimistic that the authors will be able to improve the manuscript through the revision. I recommend publication subject to the following major revisions.

General Comments

(1) Net surface heat flux anomalies associated with AD: Sea ice growth & melting rates are associated with the net surface heat flux. I suggest examining the response of net surface heat flux to the summer negative AD. In particular, is there an increasing sensitivity of net surface heat flux (more downward heat flux anomalies) to the summer negative AD? The net surface heat flux anomalies might be presented in the lower panel of Figure 7.

(2) Increasing sensitivity of sea ice cover to southerly wind strengthening: Figures 6 and 8 are the main findings of this study and these results should be explained further in detail. As the authors stated, Arctic sea ice becomes more vulnerable to the dynamical forcing such as southerly wind strengthening because of the continuous ice thinning. I recommend showing the PIOMAS ice thickness in the lower panel of Figure 8. Although PIOMAS ice thickness has large uncertainties, the general trend of ice thinning is reasonably well captured by PIOMAS.

(3) Case study: As shown in Serreze et al. (JGR 2016), I recommend examining the impact of AD on sea ice cover during the recent summers of 2016 and 2107. As noted in Serreze et al. (2016), each negative AD event has markedly different pressure and temperature patterns.

(4) Possible impacts of PDO on AD (Figures 9 & 10): The connection between PDO and AO is highly speculative. I am not sure whether these results need to be presented. I recommend deleting Figures 9 and 10 as well as Section 4 (Further Discussion).

[Figure]

Specific Comments

(5) Page 1 (lines 28-29): "Screen and Simmonds (2010) suggested the surface warming in the Arctic (a.k.a. polar amplification) plays a critical role in sea ice melting": This is not true. Screen and Simmonds (2010) suggested that diminishing sea ice has had a leading role in recent Arctic amplification.

(6) Page 6 (lines 16-17): It is difficult to tell the difference of PC time series between Fig. 3 and Fig. 5a-c. I thought these two are identical - both are PC time series of JJA mean SLP in the Arctic - am I misunderstanding? Please explain the differences more in detail.

(7) Page 6 (lines 19-24): I cannot agree with this argument. To me, there is no significant difference in the AD's SLP composites between the early and the late periods. There has been more frequent negative AD events since mid-2000's, but the individual negative AD's amplitude and pattern may not have changed much.

(8) Page 7 (lines 1-3): Rigor et al. (2002), more recently by Park et al. (2018) showed a strong relationship between winter AO and summer sea ice extent. Park, H.-S., A. L. Stewart and J.-H. Son, 2018: Dynamic and thermodynamic impacts of the winter Arctic Oscillation on summer sea ice extent. Journal of Climate, 31, 1483-1497.

(9) Page 7 (line 5): "AD in the recent period, which feature is not evident in the early period": I suggest checking grammar of this sentence.

(10) Page 7 (lines 16-17): "in order to better represent the condition for sea ice melting over far off the coast of Russia and North America": How about changing this to "to better represent the southerly wind-induced ice loss over the Pacific sector of the Arctic"?

(11) Page 7 (lines 26-29): Again, this speculative statement should be quantitatively diagnosed by calculating the net surface heat flux anomalies.

(12) Page 7 (lines 32-33): Ogi et al. (2010) did not explicitly state that the surface

wind-induced ice drift is more important than other factors. Please rephrase or delete this sentence.

(13) Page 8 (lines 4-7): I found it difficult to understand this sentence. If Figure 8 has limitation in explaining the recent changes of the AD's effect on sea ice, why is this plot presented?

(14) Page 8 (lines 10-11): I am not sure whether the outflow through the Fram Strait has recently increased. There is no obvious difference between Figs. 8a and 8b.

(15) Page 8 (lines 13-15): Figure 2b does not show any obvious changes in wind vectors around the Bring Strait.

(16) Page 9 (lines 31-32): As the authors stated, the relationship between PDO shifts and the AD center is difficult to elucidate. Again, I suggest deleting Figures 9, 10, and Section 4 (Further discussion), which is a distraction.

(17) Page 10 (line 2): "AO modulates sea ice" should be changed to "winter AO modulates sea ice". Again, more recently, Park et al. (2018) showed a nontrivial connection between the winter AO and summer sea ice.

(18) Page 10 (lines 18-19): I cannot understand this sentence. Please rephrase.

(19) Page 10 (lines 21-22): Did the authors imply "anticyclonic circulation anomalies over the Beaufort Sea"? Again, Figure 6 does not support the authors' argument.

(20) Page 10 (lines 26-27): Again, please delete this sentence.

---

## Author Comment (AC1) · 10 May 2019

**Response to Reviewer #1's Comments:**

General

This paper is telling is that September sea ice conditions are strongly shaped by atmospheric circulation patterns during summer, and that atmospheric circulation patterns are variable and have shifted over time. We have known this for many years, and a number of previous efforts have also noted that as the ice thins, relationships between atmospheric circulation anomalies and sea ice responses may be changing. In this sense, the present paper, while impressive in terms of depth of analysis, isn't really telling us anything fundamentally new.

→ As the reviewer commented out, this study basically supports the existing studies in that atmospheric circulation patterns have profound impacts on the sea ice variability in the Arctic, especially in the summer. Below we highlight our new findings with respect to the existing studies.

The Arctic Dipole (AD) mode has been known to be linked with the September sea ice extent (SIE). Wang et al. (2009) identified AO and AD as principal modes (i.e., EOF1 and EOF2, respectively) of the sea level pressure (SLP) variability in the Arctic from the analysis of the long-term data for 1948-2008, and suggested that negative AD years such as 2007 tend to show more linkage with the SIE minimum. Overland et al. (2012) also suggested that the Arctic sea ice has decreased by the series of negative AD years persistent during 2007-2012. In extending this study, Serreze et al. (2016) examined the decadal changes in the SLP patterns, and the SLP anomalies in the recent years resemble more the negative AD pattern to which the sea ice decrease was attributed. However, there was no quantitative assessment of the relationship between the sea ice extent and the AD variability was provided in the previous studies. This may be partly because the correlation between the sea ice extent and the AD index vanishes when the entire analysis period was applied since the 1980s (Fig. 5d).

What is new that we try to convince from this study is to provide a new perspective to the mechanisms responsible for the change in the relationship between SIE and AD. It is hypothesized that the principal modes may have experienced a significant change in their center of actions across the decades. In our analysis, this is particularly the case for the 2nd EOF mode (AD), although the 1st EOF mode (AO) is still predominant with no significant change in the spatial pattern. The change in the AD spatial pattern is statistically significant when the analysis period was separated before and after the late 1990s (See the statistical test result in our response to the specific comment below), and it explains why the correlation between SIE and AD is statistically significant just for the recent period (1998-2017), not in the past (1982-1997). This aspect is highlighted in detail in the manuscript based on the quantitative analysis based on the time series correlations (Fig. 5d).

This study detailed the mechanisms of how the spatial pattern change in the AD mode provides more favorable conditions for the interannual variation of the SIE, based on comprehensive analyses to the sea ice dynamic and thermodynamic fields. Among them, the sea ice dynamics associated with the low-level wind change could explain better for the sea ice variability in the recent period, rather than changes in temperature advection or heat flux from the atmosphere.

Finally, the remaining question what drives the AD pattern change in the recent decade is addressed newly in the manuscript. We highlight that the AD pattern change could appear recurrently depending on the phase of the Pacific Decadal Oscillation (PDO). Our statistical analysis based on the long-term reanalysis data (NCEP R1) dated back to 1948 proves that a similar change in the spatial pattern of AD has occurred during the negative PDO phase. We admit this is from statistics and the causal relationship could be elucidated by some numerical experiments, but this is not easy to experiment and well beyond the scope of current research.

Specific

Abstract, Page 1, Line 15: What month is this correlation based on? September ice extent against

summer circulation? Be specific.

→ The correlation is calculated between the summer Arctic Dipole index (JJA) and the September Sea ice extent, which is -0.05 in the past period and becomes 0.57 in the recent period. This information will be added in the revised manuscript.

Page 1, line 24: The downward trends in sea ice extent involve more than "rapid melting".

→ "Rapid melting" will be replaced by "radical change".

Page 1, Line 25-28: Over what period did Serreze et al. [2007] compute the trend? The 12.4% per decade trend cited in later studies is not "expedited", it is simply based on a longer sea ice record. Also, percent per decade trends are meaningless numbers without clearly citing the baseline averaging period.

→ The trend of September sea ice extent was calculated for 1979-2006 (-8.6% per decade) in Serreze et al. (2007), while Stroeve et al. (2012) used 1979-2010 (-12.6% per decade).

The sentence is somewhat misleading and it will be modified as "Based on the National Snow and Ice Data Center (NSIDC), the linear trend of the SIE during 1979-2018 relative to 1981-2010 average is -12.8 % per decade, with a more rapid declining trend in recent years". See below for the NSIDC record.

| Year of September Average Extent | Extent (million sq. km.) | Anomaly Relative to 1981-2010 Average (million sq. km.) | Anomaly Relative to 1981-2010 Average (%) | Anomaly Relative to Previous Record (million sq. km.) | Anomaly Relative to Previous Record (%) | Linear Trend Since 1979 (sq. km. per year) | Linear Trend Since 1979 Relative to 1981-2010 Average (% per decade) |
|---|---|---|---|---|---|---|---|
| 2002 | 5.83 | -0.58 | -9.1 | -0.25 | -4.1 | -45900 | -7.2 |
| 2003 | 6.12 | -0.29 | -4.6 | 0.29 | 5.0 | -47400 | -7.4 |
| 2004 | 5.98 | -0.43 | -6.8 | 0.15 | 2.6 | -49400 | -7.7 |
| 2005 | 5.50 | -0.91 | -14.2 | -0.33 | -5.7 | -54300 | -8.5 |
| 2006 | 5.86 | -0.55 | -8.6 | 0.36 | 6.5 | -55300 | -8.6 |
| 2007 | 4.27 | -2.14 | -33.4 | -1.23 | -22.4 | -66600 | -10.4 |
| 2008 | 4.69 | -1.72 | -26.9 | 0.42 | 9.8 | -72700 | -11.3 |
| 2009 | 5.26 | -1.15 | -18.0 | 0.99 | 23.2 | -73800 | -11.5 |
| 2010 | 4.87 | -1.54 | -24.1 | 0.60 | 14.1 | -76500 | -11.9 |
| 2011 | 4.56 | -1.85 | -28.9 | 0.29 | 6.8 | -79900 | -12.5 |
| 2012 | 3.57 | -2.84 | -44.3 | -0.70 | -16.4 | -87400 | -13.6 |
| 2013 | 5.21 | -1.20 | -18.8 | 1.64 | 45.9 | -85500 | -13.3 |
| 2014 | 5.22 | -1.19 | -18.6 | 1.65 | 46.2 | -83400 | -13.0 |
| 2015 | 4.62 | -1.79 | -28.0 | 1.05 | 29.4 | -83900 | -13.1 |
| 2016 | 4.53 | -1.88 | -29.4 | 0.96 | 26.9 | -84300 | -13.1 |
| 2017 | 4.82 | -1.59 | -24.8 | 1.25 | 34.0 | -83200 | -13.0 |
| 2018 | 4.71 | -1.70 | -26.6 | 1.14 | 31.9 | -82300 | -12.8 |

September Average Extents, 2002-2018: Calculated by Walt Meier and Julienne Stroeve, National Snow and Ice Data Center. All values in this table estimated based on the NSIDC Sea Ice Index Version 3. Note that these figures show September average extents rather than minimum extents.

[NSIDC]

Page 1, Line 29: Surface warming is not Arctic amplification – AA refers to a comparison between temperature trends between the Arctic and the globe as a whole (or the northern hemisphere). And there seems to be a misunderstanding here – a large component of AA seems to be due to ice loss (the ocean loses heat to the atmosphere in autumn and winter), rather than the cause of it.

→ We agree and the "Arctic Amplification" will be replaced with "global warming".

Page 2, line 3 and elsewhere in the text: A "declining trend" implies that the trend is getting smaller. The correct term is "downward trend"

→ Will be corrected as "downward trend".

Page 2, line 5: To state that the underlying mechanisms for sea ice variability in summer are still "under debate" is quite a stretch. Scientists have been looking at these mechanisms (atmospheric and oceanic variability) for many years. The authors should be citing earlier pioneering studies – from the way the text reads, there was no research on mechanisms behind sea ice variability before the dawn of the 21st century. We all stand on the shoulders of those before us. Give credit where credit is due.

→ We just tone down the text as " … are suggested with on a variety of mechanisms". We will include early pioneering studies as in the below.

Thorndike, A. S., & Colony, R. (1982). Sea ice motion in response to geostrophic winds. Journal of Geophysical Research: Oceans, 87(C8), 5845-5852.

Curry, J. A., Schramm, J. L., & Ebert, E. E. (1995). Sea ice-albedo climate feedback mechanism. Journal of Climate, 8(2), 240-247.

Parkinson, C. L., Cavalieri, D. J., Gloersen, P., Zwally, H. J., & Comiso, J. C. (1999). Arctic sea ice extents, areas, and trends, 1978–1996. Journal of Geophysical Research: Oceans, 104(C9), 20837-20856.

Page 3, lines 1-3: As I recall, Ogi et al. [2007, 2008] were discussing what called the "summer AO" pattern, not the AD.

→ Will be corrected as "summer AO".

Page 4, line 12: I leave it to another reviewer to comment on the validity of conducting an EOF analysis over such a very restricted spatial domain (70 to 90 deg. N).

→ The definition of the AD is the 2[nd] EOF mode of SLP anomalies in the area north to 70°N in Wu et al. (2006) and Watanabe et al. (2006) for winter. Wang et al. (2009) and Overland and Wang (2010) used the same domain for their summer analyses. We also found that various recent studies adopt this definition.

Wu, B., Wang, J., & Walsh, J. E. (2006). Dipole anomaly in the winter Arctic atmosphere and its association with sea ice motion. Journal of Climate, 19(2), 210-225.

Watanabe, E., Wang, J., Sumi, A., & Hasumi, H. (2006). Arctic dipole anomaly and its contribution to sea ice export from the Arctic Ocean in the 20th century. Geophysical research letters, 33(23).

Overland, J. E., & Wang, M. (2010). Large-scale atmospheric circulation changes are associated with the recent loss of Arctic sea ice. Tellus A, 62(1), 1-9.

Page 4, line 25: Again, Arctic amplification is not about the Arctic temperature trends alone, it's about the comparison between Arctic and global temperature trends. And it's not "polar amplification" - it's just the Arctic.

→ We agree and the phrase will be modified as "under global warming".

Page 5, lines 14-15. Assuming that the "Pacific section" refers to the Beaufort/Chukchi seas, why would there be a greater ice loss here in the later period when the motion is more onshore than in the previous period and would tend to transport thick ice from north of the Canadian Arctic Archipelago into the region?

→ A greater sea ice loss in this region cannot be simply explained by enhanced northerly winds toward the Beaufort Sea in the recent period as the transpolar drift stream also carries the sea ice toward the Atlantic section. Much sea ice loss and retreat in the sea ice line in the Pacific section seems to be a

consequence of various contributions such as air temperature warming, warm inflow from the Pacific, and the ice thickness change.

Page 6, line 5: As far as I can see, no trend analysis has been performed on the time series. And I don't see much of anything resembling decadal scale variability in PC1. What I see is a series of ups and downs.

→ Agreed and the sentence will be deleted in the revised manuscript.

Page 6, lines 15-25: I have a very hard time convincing myself that the patterns for the earlier and later periods shown in Figure 4 are very different. In my opinion the authors are trying to read too much into these figures.

→ We elaborate more on Fig. 4 by changing the color scheme (See below for the modified Fig. 4). Now the figure shows that the center of action tends to shift counterclockwise, and in particular the variability maximum in the western hemisphere shifted from Queen Elizabeth Islands to Greenland. To test the statistical significance, we applied the F-test for the two EOF vectors. The AD pattern change is notable over the regions of Queen Elizabeth Islands and Greenland, with the statistical significance at 5 % level (See below Fig. S1, bottom). Moreover, the pattern correlation between the two AD modes (i.e., Figs. 4c and 4d) is as low as 0.58, while that of AO (Figs. 4a and 4b) is as high as 0.99 for the area of the western hemisphere (60-90N, 0-180W). This implies that AD has experienced a significant pattern change in the recent decade, whereas AO has not.

[Figure]

**Figure 4.** The three leading EOFs of JJA-mean SLP (contour) in the early (1982-1997, left panels) and the recent (1998-2017, right) period. (a) and (b) for the first mode, (c) and (d) for the second, and (e) and (f) the third mode, respectively. The shaded area shows strong variability region of each mode. The regression pattern of surface wind anomalies (vector) is also shown in each map.

[Figure]

**Figure R1**. The difference of the leading EOFs (top: EOF 1 and bottom: EOF 2). The dotted area indicates the statistical significance at the 5 % level from the F-test. The EOF vectors were scaled by the variance represented by each mode and subject to the F-test for the variance ratio at each grid point. The degree of freedom is 15 for the early vector and 19 for the recent.

Page 7, line 16: Is melting the only thing going on here?

→ The sign of the AD vector can be reversed as the reviewer commented. The sentence will be modified as "to better represent the southerly wind-induced ice loss over the Pacific sector of the Arctic".

Page 8, line 1: The differences in ice motion between the two periods seems very nuanced to me. Again, I get the impression that that the authors are trying to read too much into the differences.

→ Following the reviewer's comment, we elaborate this part more. Figure 8 in the original manuscript will be replaced by Figure 8 shown below, in which we modify the color of the wind vector for better display. In addition, the data has been updated up to 2017 with the Sea Ice Motion version 4.

The sea ice motion associated with AD (c.f. Fig. 8a and 8b) becomes faster in the mid Arctic around the edge of the sea ice extent. In the recent period, sea ice is drifted more clearly toward the Norwegian Sea and discharged to the North Atlantic. This sea ice motion change is consistent well with the change in the surface wind driven by AD (c.f. Fig. 8c and 8d). Northerly winds have been strengthened from the Arctic to the North Atlantic in the recent period to provide a more favorable condition for sea ice to be discharged to the Atlantic.

For a better illustration of the changes in the sea ice motion and surface wind, we prepare Figure R2

below. Sea ice motion difference (Fig. R2a) shows clockwise rotation anomalies with a more enhanced transpolar drift to the Atlantic section. Although this sea ice motion change can be detected only over the sea ice covered area, corresponding surface wind change (Fig. R2b) shows the dominant feature in the downstream side where the strong outflow anomalies are found from the Arctic to the Barents Sea and to the Norwegian Sea. This study highlights that the AD pattern change provides a more favorable condition for the Arctic sea ice loss to the North Atlantic based on these analyses.

[Figure]

**Figure 8**. Regression pattern of sea ice motion (top, vector) and surface wind (bottom, vector) onto the AD index in the early (1982-1997, left) and the recent (1998-2017, right) period. Shaded is the sea ice age (top) and the sea ice thickness (bottom) in September averaged over each period.

[Figure]

**Figure R2**. Difference of (a) sea ice motion and (b) surface wind associated to the AD in each period. Shaded is difference of (a) sea ice age and (b) sea ice thickness in September averaged over each period.

Page 8, line 12: Rebecca Woodgate has a number of papers addressing links between the Bering Strait heat inflow and sea ice conditions in the Chukchi Sea and potentially beyond. Also see: 10.1002/2016JC011977, which specifically examines predictability of ice conditions in the Chukchi Sea based on the Bering Strait heat inflow.

→ As indicated, Woodgate et al. (2010) and Serreze et al. (2016) discussed the mechanisms of the warm ocean current through Bering Strait and its impacts on the sea ice in the Chukchi Sea. These studies will be added in the revised manuscript, in addition to the studies of Shimada et al. (2006) and Carmack et al. (2015).

Page 8, line 15 It needs to be acknowledge here (or somewhere ) that the last three summers have been very cyclonic over the central Arctic Ocean; in other words, the much bandied "intensification" of the Beaufort Sea high appears to have broken down.

→ We agree on the reviewer's point and it is consistent well with the mechanisms suggested in this study. Strong anticyclonic circulation over the central Arctic induced by the Beaufort Sea High tends to accelerate the transpolar sea ice drift to the Atlantic sector and provides a favorable condition for the decrease of Arctic SIE. This seems to be a dominant process particularly in 2007-2012 when the Beaufort Sea High was relatively strong (See below for Fig. 1 in Overland et al. 2012). Accordingly, the Arctic SIE exhibited below normal condition (See below Fig. R3, blue circled period). On the other hand, the last three summers have been very cyclonic over the central Arctic Ocean, to provide a less favorable condition for the transpolar drift of sea ice. Accordingly, the decline of SIE has slowed down

and above the downward trend (Fig. R3, red circled period).

[Figure]

**Figure 1.** Composite of June sea level pressure (hPa) for 2007–2012. Data are from the NCEP–NCAR Reanalysis through the NOAA/Earth Systems Research Laboratory.

[Overland et al. 2012]

[Figure]

**Figure R3.** The Arctic sea ice extent (SIE) in September in the region north of 70 N. The anomalies are the departures from the average of 1981-2010. Dashed line shows the trend before and after 1998.

Page 9, lines 16-33: I think it is very difficult to argue that the cause of the shift in the AD is due to a phase change in the PDO. All that one can really say is that the shift in the AD (which seems minor to me) is part of a large-scale pattern of change involving the PDO. The link is certainly interesting, but I'm hesitant to read too much into cause and effect

→ We basically agree with the reviewer in the point that the statistical relationship does not necessarily provide the causality between AD and PDO and it is difficult to conclude the cause of the AD shift is the phase change of PDO. This hypothesis is valid because of statistical relationship and the relevant dynamical processes might be unveiled by well-designed numerical experiments. But this is a very difficult task to experiment, having said that current state-of-the-art models are not able to reproduce realistic AD as the second EOF mode and well beyond the scope of current research.

Instead, we elaborate further on our statistical analysis. The AD mode during the negative PDO years

for 1948-2017 (Fig. 10d) resembles much the AD mode obtained during the negative years before the 1980s (Suppl. Fig. S3b). They all show the similar center of action over Greenland, and the correlation coefficient between the two is as high as 0.95 just for the area of the western hemisphere (60-90N, 0-180W). This convinces that the shift of the center of action in the AD mode is closely related to the phase of PDO.

---

## Author Comment (AC2) · 11 May 2019

**Response to Reviewer #2's Comments:**

This study examines the rapidly increasing influence of summer Arctic dipole mode (AD) on September sea ice extent over the last decade using reanalysis and observational data. The authors show that the negative AD event has been more frequent since mid-2000's (Fig. 3), and has strongly influenced September sea ice extent (Fig. 5) by decreasing sea ice cover over the Pacific sector of the Arctic (Fig. 6). The authors further present that the increasing influence of AD on summer sea ice cover is partly because of sea ice thinning (Fig. 8), which increases the sensitivity of sea ice cover to southerly winds. This study nicely expands on the work of Wang et al. (2009), but the main conclusions of this study are somewhat redundant with those of Serreze et al. (JGR 2016), which carefully analyzed the relationship between AD and summer Arctic sea ice extent. While it is a little difficult to argue that this study has enough novelty to justify publication at this stage, I think this study has great potential to become an influential paper. I am optimistic that the authors will be able to improve the manuscript through the revision. I recommend publication subject to the following major revisions.

→ We appreciate the reviewer's encouraging and constructive comments on the manuscript. As we received a similar comment regarding the novelty of this study from another reviewer, we repeat our responses in the below.

As the reviewer commented out, this study basically supports the existing studies in that atmospheric circulation patterns have profound impacts on the sea ice variability in the Arctic, especially in the summer. Below we highlight our new findings with respect to the existing studies.

The Arctic Dipole (AD) mode has been known to be linked with the September sea ice extent (SIE). Wang et al. (2009) identified AO and AD as principal modes (i.e., EOF1 and EOF2, respectively) of the sea level pressure (SLP) variability in the Arctic from the analysis of the long-term data for 1948-2008, and suggested that negative AD years such as 2007 tend to show more linkage with the SIE minimum. Overland et al. (2012) also suggested that the Arctic sea ice has decreased by the series of negative AD years persistent during 2007-2012. In extending this study, Serreze et al. (2016) examined the decadal changes in the SLP patterns, and the SLP anomalies in the recent years resemble more the negative AD pattern to which the sea ice decrease was attributed. However, there was no quantitative assessment of the relationship between the sea ice extent and the AD variability was provided in the previous studies. This may be partly because the correlation between the sea ice extent and the AD index vanishes when the entire analysis period was applied since the 1980s (Fig. 5d).

What is new that we try to convince from this study is to provide a new perspective to the mechanisms responsible for the change in the relationship between SIE and AD. It is hypothesized that the principal modes may have experienced a significant change in their center of actions across the decades. In our analysis, this is particularly the case for the 2nd EOF mode (AD), although the 1st EOF mode (AO) is still predominant with no significant change in the spatial pattern. The change in the AD spatial pattern is statistically significant when the analysis period was separated before and after the late 1990s (See the statistical test result in our response to the specific comment below), and it explains why the correlation between SIE and AD is statistically significant just for the recent period (1998-2017), not in the past (1982-1997). This aspect is highlighted in detail in the manuscript based on the quantitative analysis based on the time series correlations (Fig. 5d).

This study detailed the mechanisms of how the spatial pattern change in the AD mode provides more favorable conditions for the interannual variation of the SIE, based on comprehensive analyses to the dynamic and thermodynamic fields. Among them, the sea ice dynamics associated with the low-level wind change could explain better for the sea ice variability in the recent period, rather than changes in temperature advection or heat flux from the atmosphere.

Finally, the remaining question what drives the AD pattern change in the recent decade is addressed newly in the manuscript. We highlight that the AD pattern change could appear recurrently depending on the phase of the Pacific Decadal Oscillation (PDO). Our statistical analysis based on the long-term reanalysis data (NCEP R1) dated back to 1948 proves that a similar change in the spatial pattern of AD

has occurred during the negative PDO phase. We admit this is from statistics and the causal relationship could be elucidated by some numerical experiments, but this is not easy to experiment and well beyond the scope of current research.

General Comments

(1) Net surface heat flux anomalies associated with AD: Sea ice growth & melting rates are associated with the net surface heat flux. I suggest examining the response of net surface heat flux to the summer negative AD. In particular, is there an increasing sensitivity of net surface heat flux (more downward heat flux anomalies) to the summer negative AD? The net surface heat flux anomalies might be presented in the lower panel of Figure 7.

→ Following the reviewer's comment, we examined the net surface heat flux associated with the AD. Figure R1 below compares the net heat flux anomalies regressed onto the AD index between the past and the recent decade. Overall, the net surface heat flux anomalies are increasing in the high latitudes, the signal is less clear or even negative in the central Arctic. As this pattern does not match directly with the region of sea ice decrease, the sea ice dynamics impacted by surface wind anomalies seem to be more responsible for the sea ice variability, rather than thermodynamic processes. This aspect supports the results and conclusion in the manuscript.

We can combine these figures with the original Figure 7, as suggested, and add relevant discussion in the revised manuscript.

[Figure]

**Figure R1.** Regressed pattern of net surface heat flux onto the AD index in (a) the past and (b) the recent period. The positive values indicate downward.

(2) Increasing sensitivity of sea ice cover to southerly wind strengthening: Figures 6 and 8 are the main findings of this study and these results should be explained further in detail. As the authors stated, Arctic sea ice becomes more vulnerable to the dynamical forcing such as southerly wind strengthening because

of the continuous ice thinning. I recommend showing the PIOMAS ice thickness in the lower panel of Figure 8. Although PIOMAS ice thickness has large uncertainties, the general trend of ice thinning is reasonably well captured by PIOMAS.

→ Following the reviewer's comment, we show Figure R2 below. In the recent period, sea ice thickness becomes thin clearly, and the surface wind anomalies pass over this thin area in the edge of the Arctic sea ice extent. This sea ice thickness figure supports well our discussion with Figure 8, and we will use this figure in the revised manuscript.

[Figure]

**Figure R2.** Regressed surface wind anomalies onto the AD index and the time-mean sea ice thickness in the past and the recent period. The sea ice thickness data was obtained from the Pan-Arctic Ice Ocean Modeling and Assimilation System (PIOMAS) reanalysis by Polar Science Center.

(3) Case study: As shown in Serreze et al. (JGR 2016), I recommend examining the impact of AD on sea ice cover during the recent summers of 2016 and 2107. As noted in Serreze et al. (2016), each negative AD event has markedly different pressure and temperature patterns.

→ Serreze et al. (2016) showed the specific patterns in each year when the sea ice extent is relatively high and low, respectively. As their analysis has been done until 2015, the reviewer suggested it might be interesting to see the impact of AD on sea ice cover in recent years as a case study. Following it, we examined anomalous circulation patterns in 2016 and 2017, which are presented below in Figure R3.

Both years were featured by large negative SLP anomalies in the central Arctic and by dominant cyclonic circulation anomalies (Fig. R3), seemingly projected as typical positive AO years (See Fig. 3a and 3d in the original manuscript). Flow patterns are quite symmetric and weakly projected onto the AD mode, although both years can be classified as negative AD years (Fig. 3e). The strength of the AD mode is relatively weaker in these two years. This weak AD impact seems to be reflected in the time series of sea ice extent anomalies (Fig. 1c, see below with blue and red circles), where the downward trend of sea ice cover tends to slow down in these two years.

It is also interesting to see the difference between the two years. In 2016, SLP anomalies resemble more the negative AD mode than in 2017, with positive SLP anomalies in Greenland and negative in the eastern hemisphere. Accordingly, sea ice extent anomalies were relatively lower in 2016 than in 2017.

This supports well the dynamical mechanisms presented in this study, as well as in Serreze et al. (2016).

[Figure]

**Figure R3**. Sea level pressure and surface wind anomalies in 2016 (left) and 2017 (right) summer (JJA), respectively.

[Figure]

**Figure 1c.** The Arctic sea ice extent (SIE) in September in the region north of 70 N. The anomalies are the departures from the average of 1981-2010. Dashed line shows the trend before and after 1998.

(4) Possible impacts of PDO on AD (Figures 9 & 10): The connection between PDO and AO is highly speculative. I am not sure whether these results need to be presented. I recommend deleting Figures 9 and 10 as well as Section 4 (Further Discussion).

→ We admit the reviewer's comment that the relationship between PDO and AO is highly speculative. As we replied to the reviewer's comment in the above, without presenting numerical experiments, it is rather difficult to isolate the impacts by PDO onto the 2nd EOF mode in the Arctic SLP variability. Nevertheless, the statistical relationship between PDO and AO is quite robust and depends less on the data analysis period, suggesting a possible role of PDO onto the AD mode.

Following up this comment, we elaborate further on our statistical analysis. The AD mode during the negative PDO years for 1948-2017 (Fig. 10d) resembles much the AD mode obtained during the negative years before the 1980s (Suppl. Fig. S3b). They all show a similar center of action over

Greenland, and the correlation coefficient between the two is as high as 0.95 just for the area of the western hemisphere (60-90N, 0-180W). This convinces that the shift of the center of action in the AD mode is closely related to the phase of PDO.

In this statistical reasoning, we would like to keep this part (Section 4, Further Discussion) with Figs. 9 and 10.

Specific Comments

(5) Page 1 (lines 28-29): "Screen and Simmonds (2010) suggested the surface warming in the Arctic (a.k.a. polar amplification) plays a critical role in sea ice melting": This is not true. Screen and Simmonds (2010) suggested that diminishing sea ice has had a leading role in recent Arctic amplification.

→ We agree and the sentence will be removed.

(6) Page 6 (lines 16-17): It is difficult to tell the difference of PC time series between Fig. 3 and Fig. 5a-c. I thought these two are identical - both are PC time series of JJA mean SLP in the Arctic - am I misunderstanding? Please explain the differences more in detail.

→ The PC time series in Fig. 3d-f are repeatedly shown in Fig. 5a-c as solid lines, and they are identical. Here we present the black dashed lines in black from the "separate" EOF analysis before and after 1998. As the EOF loading patterns are not identical from the analysis with the total period and the one with a partial period, the PC timeseries are not supposed to be identical. The timeseries show much resemblance in each corresponding mode, and it suggests that each EOF modes are robustly identified as internal modes, regardless of analysis time.

(7) Page 6 (lines 19-24): I cannot agree with this argument. To me, there is no significant difference in the AD's SLP composites between the early and the late periods. There has been more frequent negative AD events since mid-2000's, but the individual negative AD's amplitude and pattern may not have changed much.

→ We elaborate more on Fig. 4 by changing the color scheme (See below for our modified version of Figure 4 in the original manuscript). Now the figure shows that the center of action tends to shift counterclockwise, and in particular the variability maximum in the western hemisphere shifted from Queen Elizabeth Islands to Greenland. To test the statistical significance, we applied the F-test for the two EOF vectors. The AD pattern change is notable over the regions of Queen Elizabeth Islands and Greenland, with the statistical significance at 5 % level (See Fig. R4 below, bottom). Moreover, the pattern correlation between the two AD modes (i.e., Figs. 4c and 4d) is as low as 0.58, while that of AO (Figs. 4a and 4b) is as high as 0.99 for the area of the western hemisphere (60-90N, 0-180W). This implies that AD has experienced a significant pattern change in the recent decade, whereas AO has not.

[Figure]

**Figure 4.** The three leading EOFs of JJA-mean SLP (contour) in the early (1982-1997, left panels) and the recent (1998-2017, right) period. (a) and (b) for the first mode, (c) and (d) for the second, and (e) and (f) the third mode, respectively. The shaded area shows strong variability region of each mode. The regression pattern of surface wind anomalies (vector) is also shown in each map.

[Figure]

**Figure R4**. The difference of the leading EOFs (top: EOF 1 and bottom: EOF 2). The dotted area indicates the statistical significance at the 5 % level from the F-test. The EOF vectors were scaled by the variance represented by each mode and subject to the F-test for the variance ratio at each grid point. The degree of freedom is 15 for the early vector and 19 for the recent.

(8) Page 7 (lines 1-3): Rigor et al. (2002), more recently by Park et al. (2018) showed a strong relationship between winter AO and summer sea ice extent. Park, H.-S., A. L. Stewart and J.-H. Son, 2018: Dynamic and thermodynamic impacts of the winter Arctic Oscillation on summer sea ice extent. Journal of Climate, 31, 1483-1497.

→ Park et al. (2018) will be added in the revised manuscript. We will also briefly discuss the results from Park et al. (2008), which highlight the connection between wintertime AO circulation anomalies on the following summer sea ice extent.

(9) Page 7 (line 5): "AD in the recent period, which feature is not evident in the early period": I suggest checking grammar of this sentence.

→ Will be modified as "….AD in the recent period. This feature is not evident in the early period".

(10) Page 7 (lines 16-17): "in order to better represent the condition for sea ice melting over far off the coast of Russia and North America": How about changing this to "to better represent the southerly wind-induced ice loss over the Pacific sector of the Arctic"?

→ The sentence will be changed as suggested.

(11) Page 7 (lines 26-29): Again, this speculative statement should be quantitatively diagnosed by calculating the net surface heat flux anomalies.

→ It was rather difficult to find the direct relationship between the net surface heat flux anomalies and the sea ice cover changes in response to the AD change. See our reply to the reviewer's specific comment (1) in the above.

(12) Page 7 (lines 32-33): Ogi et al. (2010) did not explicitly state that the surface wind-induced ice drift is more important than other factors. Please rephrase or delete this sentence.

→ Agreed and the reference will be removed in the sentence.

(13) Page 8 (lines 4-7): I found it difficult to understand this sentence. If Figure 8 has limitation in explaining the recent changes of the AD's effect on sea ice, why is this plot presented?

→ The analysis of the sea ice motion was conducted to examine its strong relationship with the surface wind in the Arctic. Even though the sea ice motion can be detected only over the ice-covered region, it shows good correspondence with the surface wind anomalies. Due to limitation in understanding sea ice motion, Figure 8 in the original manuscript will be replaced by Figure 8 shown below which includes surface wind pattern.

(14) Page 8 (lines 10-11): I am not sure whether the outflow through the Fram Strait has recently increased. There is no obvious difference between Figs. 8a and 8b.

→ Following up the reviewer's comment, we elaborate this part more. Figure 8 in the original manuscript will be replaced by Figure 8 shown below, in which we modify the color of the vector for better display. In addition, the data has been updated up to 2017 with the Sea Ice Motion version 4.

The sea ice motion associated with AD (c.f. Fig. 8a and 8b) becomes faster in the mid Arctic around the edge of the sea ice extent. In the recent period, sea ice is drifted more clearly toward the Norwegian Sea and discharged to the North Atlantic. This sea ice motion change is consistent well with the change in the surface wind driven by AD (c.f. Fig. 8c and 8d). Northerly winds have been strengthened from the Arctic to the North Atlantic in the recent period to provide a more favorable condition for sea ice to be discharged to the Atlantic.

For a better illustration of the changes in the sea ice motion and surface wind, we prepare Figure R5 below. Sea ice motion difference (Fig. R5a) shows clockwise rotation anomalies with a more enhanced transpolar drift to the Atlantic section. Although this sea ice motion change can be detected only over the sea ice covered area, corresponding surface wind change (Fig. R5b) shows the dominant feature in the downstream side where the strong outflow anomalies are found from the Arctic to the Barents Sea and to the Norwegian Sea. This study highlights that the AD pattern change provides a more favorable condition for the Arctic sea ice loss to the North Atlantic based on these analyses.

[Figure]

**Figure 8**. Regression pattern of sea ice motion (top, vector) and surface wind (bottom, vector) onto the AD index in the early (1982-1997, left) and the recent (1998-2017, right) period. Shaded is the sea ice age (top) and the sea ice thickness (bottom) in September averaged over each period.

[Figure]

**Figure R5**. Difference of (a) sea ice motion and (b) surface wind associated to the AD in each period. Shaded is difference of (a) sea ice age and (b) sea ice thickness in September averaged over each period.

(15) Page 8 (lines 13-15): Figure 2b does not show any obvious changes in wind vectors around the Bring Strait.

→ It was mistyped and will be corrected as "Figure 6". Enhanced easterlies over the Chukchi Sea is related to the Ekman transport of warm oceanic inflow through Bering Strait.

(16) Page 9 (lines 31-32): As the authors stated, the relationship between PDO shifts and the AD center is difficult to elucidate. Again, I suggest deleting Figures 9, 10, and Section 4 (Further discussion), which is a distraction.

→ Please see our response in the above to the comment (4).

(17) Page 10 (line 2): "AO modulates sea ice" should be changed to "winter AO modulates sea ice". Again, more recently, Park et al. (2018) showed a nontrivial connection between the winter AO and summer sea ice.

→ Will be corrected as suggested.

(18) Page 10 (lines 18-19): I cannot understand this sentence. Please rephrase.

→ We wanted to suggest that warmer temperature anomalies in the recent period (Fig. 7b) associated with AD could drive less accumulation of sea ice in the western hemisphere and provides a more favorable condition for sea ice outflow to the Atlantic. As much speculative, this sentence will be removed.

(19) Page 10 (lines 21-22): Did the authors imply "anticyclonic circulation anomalies over the Beaufort Sea"? Again, Figure 6 does not support the authors' argument.

→ "Cyclonic" circulation is a typo and it will be corrected as "anti-cyclonic" circulation. Strong Beaufort High might drive oceanic inflow through the Bering Strait via Ekman Transport.

(20) Page 10 (lines 26-27): Again, please delete this sentence.

→ Please see our response to the reviewer's general comment and the specific comment in (4).

---

## Referee Report (RR1)

I've found that the first author has carefully responded to all of my questions in detail. I would like to thank the first author for his/her efforts for improving the manuscript. In particular, Figure S3, Figure R2 and Figure S4 are key figures that can nicely explain the different sea ice responses to the summer AD during the early vs. recent periods. Therefore, these plots need to appear as main figures and need to be illustrated more in detail. I am surprised to see that these nice plots (Figures S3, S4, R2) are in the supplementary information (or just for response to the reviewer).

I recommend publication subject to the following minor revision, but they are mandatory. I am willing to review the revised manuscript once more.

(1) Mechanisms: I cannot understand why it is important to show Figure 7 (surface air temperature anomalies associated with AD during the early and the recent periods). Instead, Figures S3, S4 (supp. Fig 3 and Fig 4) and R2 (response Fig 2) can nicely illustrate the mechanisms of sea ice loss during the summer AD events. These plots, Figures S3, S4 and R2 need to appear as main figures in the main text and need to be illustrated more in detail. Instead, I recommend moving the current Figure 7 (SAT anomalies) to supplementary information.

(2) In particular, Figure R2 nicely presents the intensification of Beaufort Gyre over the last decade. Please see Zhang et al. (2016). "The Beaufort Gyre intensification and stabilization: A model-observation synthesis". I strongly recommend introducing Zhang et al (2016) and explaining Figure R2 more in detail.

(3) Shortwave radiation (page 8, lines 19 – 20): "The surface heat flux anomalies are mostly contributed by the changes in the shortwave radiation terms (not shown)"

This result should be shown as a main figure and should be explained more in detail.

(4) Surface Albedo changes (page 8, lines 21 – 22): "The response of surface albedo becomes larger and much clearer.."

This is a main finding of this study and needs to be emphasized more – I strongly recommend

illustrating this finding in the Abstract.

(5) OLR (page 8, lines 11 – 14): "It is also found that the regressed OLR.. increase (decrease) of solar radiation".

It is difficult to follow what the authors are trying to say. Do the authors argue that the increased OLR (less clouds) can melt more sea ice by increasing downward shortwave radiation at the surface? The increased downward shortwave radiation is usually compensated by the decreased downward longwave radiation. So, it is difficult to conclude that the increased OLR can melt more sea ice in the summer.

(6) Pattern correlation (page 7, lines 1 – 4): I appreciate adding this explanation. However, I suggest adding this calculation (r=0.76 as 95% confidence interval) in the supplementary information. Also, there is no illustration on the spatial pattern correlation of AD between the early and the recent periods: in the previous manuscript, the pattern correlation between Fig. 4c and 4d was written as 0.58, but in this revised version the correlation coefficient is missing. The authors may have accidently deleted this illustration.

(7) Page 7 line 20: "although the persistence of the observed AO index is too short to connect the winter AO with the summer sea ice extent"

I cannot understand this analogy. Williams et al (2016) emphasized the importance of AO-induced precondition not the persistence of seasonal AO.

(8) Color bars of Figure 2a and Figure 6: I suggest changing the color of Figure 2a and Figure 6 to reds (warm colors).

(9) The impact of transpolar drift on summer sea ice loss (page 2 lines 14 – 24): Park and Stewart (TC 2016) also showed that the southerly wind strengthening can effectively decrease summer sea ice cover over the Pacific sector of the Arctic.

Park, H.-S. and A. L. Stewart, 2016: An analytical model for wind-driven Arctic summer sea ice drift, The Cryosphere, 10, 227-244.

(10) Summery and Conclusion: This section need to be rewritten after adding Figures S3, S4 and R2 as main figures and after revising their illustrations on the mechanisms.

---

## Author Response (AR2)

**Response to the Review's Comments**

The manuscript has been revised carefully according to the reviewer's constructive comments. We believe the manuscript is further improved by more clarifications, particularly for the dynamical and thermodynamical processes engaged with the AD variability. Below is our line-by-line response to the reviewer's specific comments.

(1) Calculating the net surface heat flux anomalies associated with the summer AD
The decrease in upward shortwave radiation (decreased albedo) is usually a dominant factor for the net surface heat flux (SW_up + SW_dn + LW_up + LW_dn + SHF + LHF). I cannot understand why the authors were not able to see the decreased net surface heat flux over the regions where SIC decreased. I would urge the authors to re-calculate the net surface heat flux anomalies during the summers of strong negative AD. The authors should be able to see a significant decrease in upward shortwave radiation at the surface (SW_up) over the regions where SIC is anomalously small even though the SIC reduction was primarily driven by winds. Also, the net surface heat flux anomalies over the land need not be plotted because we are focusing on the heat flux over the Arctic Ocean.

➜ Following the reviewer's comment, we modified Figure S3 by showing the total net surface heat flux anomalies associated with the summer AD including radiative flux terms. We also masked out the values over the land.

➜ As the reviewer surmised, the surface heat flux change driven by AD (Fig. S3) corresponds well with the sea ice concentration change pattern shown in Fig. 6. The net downward surface heat flux anomalies are negative in the sea ice increasing region such as in the Atlantic sector of the Arctic Sea, and positive for the rest of the regions where the sea ice decreases. The surface heat flux anomalies are mostly contributed by the changes in the shortwave radiation terms (See Fig. R1 below for the comparison of each contributing terms to the total surface heat flux). In particular, the upward shortwave radiation (Fig. R1e) tends to decrease significantly in the East Siberia, Chukchi, and Beaufort Seas where the sea ice melting signal by AD is pronounced in the recent period (Fig. 6b). This process is also reflected in the reduction of surface albedo due to the sea ice decrease (Fig. R2). The response of surface albedo becomes larger and much clearer in the recent period.

➜ As the reviewer commented out, aforementioned changes in surface flux associated with the summer AD suggest that the sea ice-albedo feedback may also contribute to the changes in sea ice concentration in addition to the transpolar sea ice drift by AD wind anomalies. Therefore, one cannot rule out the important role of the thermodynamical process for the sea ice variability associated with AD.

➜ We reflected this point in the revised manuscript (Page 8 line 14-25).

[Figure]

**Figure S3**. Regression patterns of net downward surface heat flux (sensible + latent + net short wave + net long wave, W m-2) onto the AD index for (a) the early (1982-1997) and (b) the recent (1998-2017) period. The heat flux is defined as positive for the downward, and the AD index is reversed in sign before regression. Dotted area indicates the statistically significant region at the 95% confidence level.

[Figure]

**Figure 6.** Regression patterns of sea ice concentration (shaded, %) and the surface wind anomalies (vector, m s-1) onto the AD index for (a) the early (1982-1997) and (b) the recent (1998-2017) period. The AD index was reversed in sign for the melting phase of sea ice. Colored lines indicate the time-mean sea ice concentration of 15 % (red dotted), 30 % (red dashed), 50 % (red solid) and 80 % (blue solid) in each period

[Figure]

**Figure R1.** Surface heat flux changes associated with the summer AD in the recent (1997-2017) period. Sensible and latent heat fluxes are defined as positive for the upward. The AD index is reversed in sign before regression.

[Figure]

**Figure R2.** Surface albedo changes associated with the summer AD in (a) the past (1982-1997) and (b) the recent (1997-2017) periods, respectively. The AD index is reversed in sign before regression.

(2) Calculating the ice flux divergence using reanalysis data such as PIOMAS and ORAP-5

The authors should be able to calculate the wind-driven ice flux divergence anomalies associated with the summer AD using reanalysis data such as PIOMAS and ORAP-5. I understand that this study attempts to illustrate the processes using observations, but the suggested processes – the increased impact of summer AD on sea ice (Figures 5 and 6)– are not highly original as pointed out by the other Reviewer. Therefore, more quantitative analyses are required to justify publication.

➔ Following the reviewer's comment, we tried to calculate the ice flux divergence using PIOMAS data. In Park et al. (2018), the ice flux divergence is represented as:

$$\frac{\partial h}{\partial t} = -\left[\frac{\partial}{\partial x}(uh) + \frac{\partial}{\partial y}(vh)\right],$$

where h is sea ice thickness, and u and v are the ice drift in zonal and meridional direction, respectively.

Park, H. S., Stewart, A. L., & Son, J. H. (2018). Dynamic and thermodynamic impacts of the winter Arctic Oscillation on summer sea ice extent. Journal of Climate, 31(4), 1483-1497.

➔ Figure R3 shows the regressed ice flux divergence pattern associated with the summer AD. Comparing the recent and past period, the sea ice convergence is dominant over the Fram Strait in the past period, whereas the ice flux convergence becomes much weaker in the recent period. Overall, the ice flux convergence pattern basically shows the contrast between the Pacific sector (ice flux divergence) and the Atlantic sector (convergence) driven by AD, regardless of periods. However the signal becomes weaker and disorganized in the recent period, and we speculate for the reason the sea ice thickness must have decreased in the Pacific sector significantly in the recent period. In fact, the climatological sea ice thickness is less than 1.5 m over this region. Therefore we conclude the PIOMAS is not adequate for the quantitative analysis to highlight the mechanisms suggested in this study.

➔ Instead of ice flux convergence using PIOMAS reanalysis, we calculated the convergence of sea

ice motion obtained from the Polar Pathfinder daily gridded sea ice motion data (Fig. S4). The convergence of sea ice motion is defined as:

$$-\nabla \cdot V = -\left(\frac{\partial}{\partial x}u + \frac{\partial}{\partial y}v\right)$$

where u and v are the sea ice motion.

➔ The result shows that the sea ice divergence anomalies associated with the AD becomes clear in the Chukchi and the Beaufort Seas due to more enhanced transpolar sea ice drift motion toward the Atlantic sector.

➔ We add the discussion and the supplementary figure in the revised manuscript (Page 8, line 33-Page 9, line 2).

[Figure]

**Figure R3.** Regressed summer sea ice flux convergence (JJA) associated with the summer AD in (a) the early (1982-1997) and (b) the recent (1998-2017) period, respectively. Sea ice flux is obtained from PIOMAS data (Park et al. 2018).

[Figure]

**Figure S4.** Regressed sea ice motion vector (arrow, cm s$^{-1}$) and its convergence (shaded, 100 day$^{-1}$) onto the AD index in (a) the early (1982-1997) and (b) the recent (1998-2017) period, respectively.

(3) Pattern correlation between Fig. 4c and 4d

I still cannot see any significant difference in the AD's SLP pattern before (Fig. 4c) and after 1997 (Fig. 4d). More explanations are needed on the pattern correlation. While the pattern correlation coefficient 0.58 is not low, the authors argue that this value verifies a statistically significant difference between these two SLP patterns, which I cannot understand.

➔ As the atmospheric circulation anomalies are varying in large spatial scale, each grid point is not regarded as an independent sample. According to Bretherton et al. (1999), the statistical significance test should be based on the effective sample size, which can be calculated as

$$N^* = N \ \frac{1 - r_1 r_2}{1 + r_1 r_2},$$

N is the total grid size, r1 and r2 is the autocorrelation applied to the spatial pattern shifted by one grid for the early and the recent pattern, respectively.

Bretherton, C. S., Widmann, M., Dymnikov, V. P., Wallace, J. M., & Bladé, I. (1999). The effective number of spatial degrees of freedom of a time-varying field. Journal of climate, 12(7), 1990-2009.

➔ In the western hemisphere, the total sample size for pattern correlation is 17,620 (61 lat. x 289 lon.) and the effective sample size is much reduced as 6.05 (~ 7), due to the high coherency in the spatial pattern. When the sample size is 7, the 95% confidence level for the similarity should be above 0.76. The correlation coefficient of 0.58 in AD is not significant at the 90% confidence level. While the correlation coefficient of 0.99 in AO is significant at the 99% confidence level.

➔ We documented this in the revised manuscript (Page 6, line 33 – Page 7, line 4). This result is also consistent with our previous F-test results (See our previous Authors' Comment to this issue).

(4) Page 7 (lines 17 – 19): "the strong relationship with SIE is largely due to the change in the AD in the recent period"

It is difficult to understand how the authors could draw such as strong conclusion from the correlation coefficient.

➔ As the reviewer pointed out, the statement is merely based on the statistical relationship, and the responsible mechanism studies are followed in the next subsection. We tone down the statement in the revised manuscript (Page 7, line 20-23)

(5) Page 8 (lines 1 – 2): Again, Figure 6 is the main finding of this study, but comprehensive mechanisms were not suggested by the authors. Why are the recent ADs far more efficient in decreasing the SIC over the Pacific sector of the Arctic? I understand this is not an easy question to answer, but the authors need to provide mechanisms for this. I recommend utilizing sea ice model outputs such as PIOMAS and ORAP-5. For example, PIOMAS provides sea ice thickness changes associated with the ice flux divergence.

➔ Regarding the reason why the recent ADs are far more efficient in decreasing SIC, we tried to convince the readers based on the following processes that we highlighted in the revised manuscript. The relevant text is carefully revised for clarification (Page 9, 1st and 3rd para)

1) Sea ice discharge is much effective in the recent period in the Atlantic sector which can reduce the upstream of sea ice more effectively (Fig. 8, Page 9 line 3-8).

2)  The surface wind over the Chukchi Sea is changed from meridional to zonal wind in the recent period. The zonal surface wind seems to transport warm pacific water by Ekman transport, which is also related to the Beaufort High suggested by Wu et al. (2014) (Page 9 line 9-20).

3)  Thinner sea ice thickness makes sea ice become more vulnerable to the dynamical forcing (Page 9 line 21-24).

➔  We agree with the reviewer that most processes remain speculative. The responsible mechanisms should be better tested with the numerical simulation or the analysis of the sea ice model outputs. Although we attempted with the sea ice model outputs from PIOMAS, the results turned out to be less clear to support the proposed mechanisms in this study, presumably due to large uncertainty in the numerical models with less constraint from observations. Suggested modeling study or further analysis seems to be well beyond the scope of the current study and it will be pursued in the follow-up studies.

(6) Page 8 (lines 15 – 17): I cannot see any clear difference in ice drift speed in the central Arctic between these two periods (Figs 8a and 8b).

➔  We now add the difference map for sea ice motion vector in Fig. 8c. The sea ice drift becomes much stronger in the recent period over the circled region in the figure below (Page 8 line 28-29).

[Figure]

**Figure R4.** Difference of sea ice motion associated to the AD in each period. Shaded is difference of sea ice age in September averaged over each period.

(7) Page 9 (lines 19 – 27)
The suggested mechanism is highly speculative. I cannot understand why the anomalously low surface air temperature in the Atlantic sector of the Arctic is such an important factor for the wind-driven ice flux divergence over the Pacific sector of the Arctic.

➔  Agreed on the reviewer's comment, we removed our speculative explanation on the role of low surface air temperature. The whole paragraph was rewritten in the revised manuscript (Page 10, line

1-11).

---

## Author Response (AR3)

**Response to Reviewer's Comments:**

I've found that the first author has carefully responded to all of my questions in detail. I would like to thank the first author for his/her efforts for improving the manuscript. In particular, Figure S3, Figure R2 and Figure S4 are key figures that can nicely explain the different sea ice responses to the summer AD during the early vs. recent periods. Therefore, these plots need to appear as main figures and need to be illustrated more in detail. I am surprised to see that these nice plots (Figures S3, S4, R2) are in the supplementary information (or just for response to the reviewer).

I recommend publication subject to the following minor revision, but they are mandatory. I am willing to review the revised manuscript once more.

We appreciate additional comments from the reviewer. We reflect all the comments from the reviewer and please find our line-by-line responses below.

(1) Mechanisms: I cannot understand why it is important to show Figure 7 (surface air temperature anomalies associated with AD during the early and the recent periods). Instead, Figures S3, S4 (supp. Fig 3 and Fig 4) and R2 (response Fig 2) can nicely illustrate the mechanisms of sea ice loss during the summer AD events. These plots, Figures S3, S4 and R2 need to appear as main figures in the main text and need to be illustrated more in detail. Instead, I recommend moving the current Figure 7 (SAT anomalies) to supplementary information.

→ As suggested, we move Figs. S3 and S4 to Figs. 7 and 10, respectively. In addition, Fig. R2 becomes Fig. 8 in the revised manuscript. Old Fig. 7 has been moved to the supplementary figure as in Fig. S1.

→ The text has been modified to reflect these changes (P5, L5-7; P8, L14-26; P9, L6).

(2) In particular, Figure R2 nicely presents the intensification of Beaufort Gyre over the last decade. Please see Zhang et al. (2016). "The Beaufort Gyre intensification and stabilization: A model-observation synthesis". I strongly recommend introducing Zhang et al (2016) and explaining Figure R2 more in detail.

→ Yes, the surface albedo decrease in the Beaufort Sea seems to be related with the intensification of the Beaufort High and the Beaufort Gyre.

→ As the study of Zhang et al. (2016) regarding the intensification of the Beaufort Gyre is better connected with Figure 2b showing the intensification of anticyclonic circulation anomalies in the Beaufort Sea, we introduced their study after Fig. 2b (P6, L2-3).

(3) Shortwave radiation (page 8, lines 19 – 20): "The surface heat flux anomalies are mostly contributed by the changes in the shortwave radiation terms (not shown)"

This result should be shown as a main figure and should be explained more in detail.

→ Following the reviewer's comment in 1), we also add the albedo change pattern in Fig. 8, which also demonstrates the importance of the shortwave radiation change. Avoiding redundancy in the text, we decide to show the figures for each flux term change in the supplementary material (P8, L23; Supplementary Fig. S3).

(4) Surface Albedo changes (page 8, lines 21 – 22): "The response of surface albedo becomes larger and much clearer.."

This is a main finding of this study and needs to be emphasized more – I strongly recommend illustrating this finding in the Abstract.

→ We add it in the abstract (P1, L19-21).

(5) OLR (page 8, lines 11 – 14): "It is also found that the regressed OLR.. increase (decrease) of solar radiation".

It is difficult to follow what the authors are trying to say. Do the authors argue that the increased OLR (less clouds) can melt more sea ice by increasing downward shortwave radiation at the surface? The increased downward shortwave radiation is usually compensated by the decreased downward longwave radiation. So, it is difficult to conclude that the increased OLR can melt more sea ice in the summer.

→ We originally suggest anticyclonic circulation anomalies tend to decrease cloudiness and thereby increase net downward shortwave and OLR. Much resemblance in the regression pattern between OLR and 850-hPa temperature indicates this relationship. However, we do not carefully separate the cloud radiative effects in the reanalysis between shortwave and longwave radiation, because existing reanalyses such as in the NCEP/NCAR R1 data used in this study may contain large uncertainty in the estimates. Therefore, we discard the relevant discussion (P8, L20).

(6) Pattern correlation (page 7, lines 1 – 4): I appreciate adding this explanation. However, I suggest adding this calculation (r=0.76 as 95% confidence interval) in the supplementary information. Also, there is no illustration on the spatial pattern correlation of AD between the early and the recent periods: in the previous manuscript, the pattern correlation between Fig. 4c and 4d was written as 0.58, but in this revised version the correlation coefficient is missing. The authors may have accidently deleted this illustration.

→ Following the reviewer's comment, we moved the explanation to the supplementary material. We also indicate the value (r=0.58) in the main text (P7, L11-14).

(7) Page 7 line 20: "although the persistence of the observed AO index is too short to connect the winter AO with the summer sea ice extent"

I cannot understand this analogy. Williams et al (2016) emphasized the importance of AO- induced precondition not the persistence of seasonal AO.

→ Agreed and removed the phrase in the text (P7, L28).

(8) Color bars of Figure 2a and Figure 6: I suggest changing the color of Figure 2a and Figure 6 to reds (warm colors).

→ Modified as suggested (Figs. 2a, 6, 8c, and 8f).

(9) The impact of transpolar drift on summer sea ice loss (page 2 lines 14 – 24): Park and Stewart (TC 2016) also showed that the southerly wind strengthening can effectively decrease summer sea ice cover over the Pacific sector of the Arctic.

Park, H.-S. and A. L. Stewart, 2016: An analytical model for wind-driven Arctic summer sea ice drift, The Cryosphere, 10, 227-244.

→ The paper is referenced in the main text (P2, L20-22).

(10) Summery and Conclusion: This section need to be rewritten after adding Figures S3, S4 and R2 as main figures and after revising their illustrations on the mechanisms.

→ We revised the summary and conclusion (P11, L24-27).

-END-

---

## Author Response (AR4)

**Response to Reviewer's Comments:**

Figure 8 (surface albedo): I think the color bar should be flipped for consistency with Figure 7. In Figure 7, net downward heat fluxes are shaded as warm colors.

→ Modified as suggested (Fig 8).

-END-